# Dynamic LLM-Agent Network: An LLM-agent Collaboration Framework with Agent Team Optimization

## Abstract

Large language model (LLM) agents have been shown effective on a wide range of tasks, and by ensembling multiple LLM agents, their performances could be further improved. Existing approaches employ a fixed set of agents to interact with each other in a static architecture, which limits their generalizability to various tasks and requires strong human prior in designing these agents. In this work, we propose to construct a strategic team of agents communicating in a dynamic interaction architecture based on the task query. Specifically, we build a framework named Dynamic LLM-Agent Network (**DyLAN**) for LLM-agent collaboration on complicated tasks like reasoning and code generation. DyLAN enables agents to interact for multiple rounds in a dynamic architecture with inference-time agent selection and an early-stopping mechanism to improve performance and efficiency. We further design an automatic agent team optimization algorithm based on an unsupervised metric termed *Agent Importance Score*, enabling the selection of best agents based on the contribution each agent makes. Empirically, we demonstrate that DyLAN performs well in both reasoning and code generation tasks with reasonable computational cost. DyLAN achieves 13.0% and 13.3% improvement on MATH and HumanEval, respectively, compared to a single execution on GPT-35-turbo. On specific subjects of MMLU, agent team optimization in DyLAN increases accuracy by up to 25.0%. [1]

## 1 Introduction

Large language model (LLM) agents (Richards & et al., 2023; Nakajima, 2023; Reworkd, 2023) have achieved promising performance on a wide range of tasks, ranging from reasoning (Yao et al., 2023) to code generation (Shinn et al., 2023), and even to embodied tasks such as video gaming (Wang et al., 2023) and autopilot (Jin et al., 2023). Recent works tackle complicated tasks by combining different LLM agents in a team to collaborate on the same query (Li et al., 2023; Du et al., 2023; Wang et al., 2023b; Jiang et al., 2023; Shinn et al., 2023; Zheng et al., 2023; Wu et al., 2023). In these works, such LLM-agent team is often set up statically with predefined roles for each agent, such as introducing programmer and tester to communicate sequentially in a fixed order to improve code generation (Dong et al., 2023).

In the case of collaborative work between different LLM agents on a specific task, a static setup with predefined roles can, however, have several problems: (1) Manual setups require us to design task-specific roles for LLM agents in each specific domain (Dong et al., 2023), making it difficult to generalize to other domains or distinct tasks. (2) Additionally, under a static setup, LLM agents interact to produce answers in a fixed order (Shinn et al., 2023) or in one single round (Jiang et al., 2023), which might result in sensitivity to the domain and complexity of tasks. (3) Predefined roles require strong human priors to design, and may not well align with the actual situations. In addition, there is no systematic way to ensure that the LLM-agent collaboration systems have a sufficient number of LLM agents and, most importantly, an optimized team of agents.

---

[1] We will release the code once the paper is accepted.

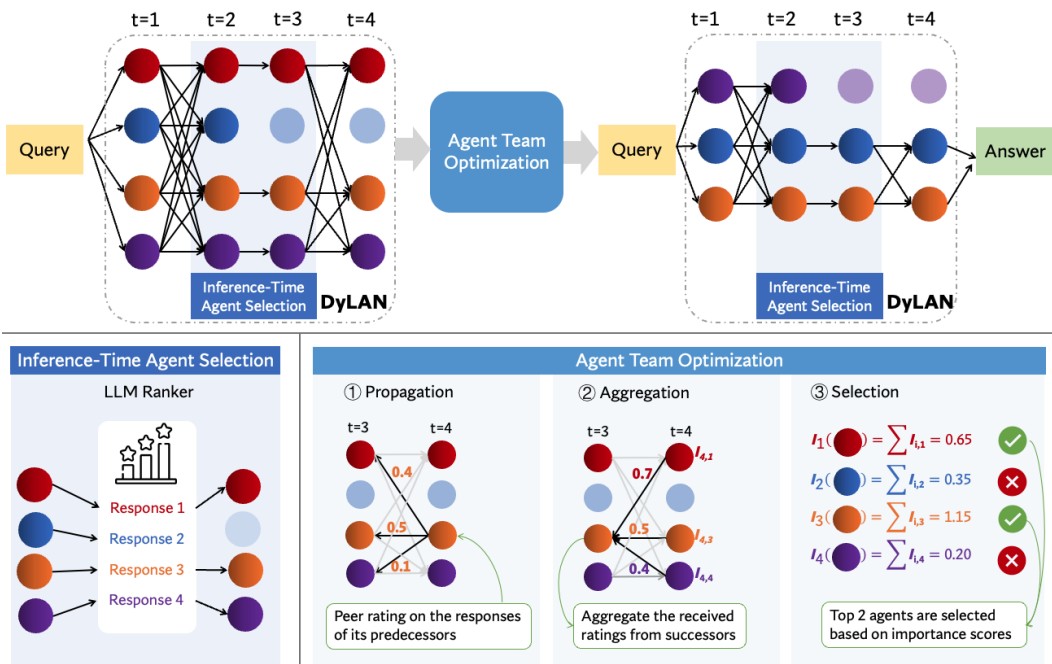

Figure 1: Overview of DyLAN. The top part shows how DyLAN outputs the answer in a feed-forward manner (Section 3). Inference-time agent selection is implemented with an LLM-empowered ranker (bottom-left), during which the low-performing agent is deactivated in subsequent time steps. In agent team optimization (Section 3.3), first the contribution of each agent is automatically evaluated using *Agent Importance Score*, denoted as $I$. Then the top-ranked agents based on $I$ will be selected as the optimized team of agents (bottom-right).

The landscape of LLM-agent collaborations requires a systematic framework in order to improve generalizability, efficiency, and performance. Therefore, several properties should be exhibited: (1) *Task Agnostic*: Prior works have suffered from the difficulty of generalizing across various domains due to the dependence on task-specific tools. A task-agnostic system can facilitate the fast adaptation of existing approaches to new situations. (2) *Efficient*: Instead of assigning agents in a static pattern, dynamically removing agents with uninformative responses can prevent the creation of useless information as well as ensure accuracy in the process of reaching consensus. (3) *Agent Team Optimization*: With thousands of open-source and unlimited LLM-generated prompts that serve a variety of roles, it is difficult to identify what the optimal team of agents might be. Ideally, multi-agent systems should be able to adapt their composition in response to the particular domain of a query with minimal supervision. While a few efforts have been made (Chen et al., 2023b; Ruan et al., 2023; Zhang et al., 2023; Wang et al., 2023b; Lu et al., 2023; Aggarwal et al., 2023; Liu et al., 2023b; Besta et al., 2023), a framework that integrates all of these aspects is still lacking.

To fill these gaps, in this work, we introduce a framework named Dynamic LLM-Agent Network (**DyLAN**) in Section 3. Concretely, we employ a feed-forward network to formulate the process of LLM-agent collaboration for arbitrary tasks. As a result, we view LLM agents at specific time steps as nodes in a network and the messages they exchange at different time steps as edges. In this way, we can organize multi-round LLM-agent collaboration into a multilayered network. To enhance the efficiency and the performance of LLM-agent collaboration on various tasks, we use an LLM-empowered ranker (Qin et al., 2023) to rank different LLM agents and deactivate low-performing agents in the subsequent interaction (i.e., inference-time agent selection), thereby creating a dynamic architecture of interactions. Additionally, we introduce an early-stopping mechanism via a Byzantine Consensus to ensure efficiency, by terminating the inference process when agents within the same layer reach consensus. Moreover, we propose automatic agent team optimization without strong human priors to increase the efficiency and effectiveness of agent team initialization. Specifically, to select the top-$k$ agents, we introduce a three-step procedure where we first ask each agent to rate its predecessors on their solutions (*propagation*), and then for each agent, *aggregate*

the ratings from their successors to quantify its contribution (*aggregation*). Finally, after summing up the ratings across all time steps, we derive an *Agent Importance Score* for each agent. Then, we can select the top-performing agents based on their importance scores to obtain the optimized team of agents (*selection*). In this way, DyLAN achieves the task-agnostic property by formulating LLM-agent collaboration into a feed-forward network to decouple the interaction architecture and task-specific design, exhibits efficiency through inference-time agent selection and early-stopping mechanism, and enables agent team optimization through a selection algorithm based on agent importance scores. We evaluate DyLAN on multiple representative tasks such as general reasoning, arithmetic reasoning, and code generation, and find that DyLAN demonstrates higher accuracy and efficiency over baselines. We also show that *Agent Importance Score* can be used as a solid unsupervised indicator for optimizing the composition of agents for DyLAN. On specific subjects of MMLU, agent team optimization in DyLAN increases accuracy by up to 25.0%.

To sum up, our contributions are as follows:

- We propose a generic LLM-agent collaboration framework DyLAN that organizes agents into a multi-layered feed-forward network with dynamic architecture by introducing inference-time agent selection and early-stopping mechanism.
- We develop an automatic agent team optimization algorithm for DyLAN based on an unsupervised *Agent Importance Score*, in order to achieve both efficiency and effectiveness.
- Empirically, DyLAN demonstrates high accuracy, efficiency, and stability in general reasoning, arithmetic reasoning, and code generation tasks.

## 2 RELATED WORK

**Interaction Architecture in LLM-agent Collaboration**   Collaboration between multiple LLM agents has demonstrated strong performance on a variety of tasks in recent years and has emerged as a promising approach to enhance the capabilities of individual LLMs. To enable collaborations between multiple agents, recent studies have developed different interaction architectures and assigned agents in static patterns. For instance, Du et al. (2023); Liang et al. (2023); Xiong et al. (2023) take multiple LLM instances into a debate for a fixed number of rounds, thus boosting their factuality and reasoning capacities. Instead of calling LLMs iteratively, Ning et al. (2023) distribute LLMs in parallel and concatenates their answers to produce better results. To aggregate multiple LLM responses, Jiang et al. (2023) generate candidates by different LLMs in one round and uses pairwise ranking to combine the top responses into one final response. It is worth noting that Hao et al. (2023) organizes LLM instances into linear layers and adopts supervised learning in context space, not the scenario in which we are interested. However, running LLMs in a static architecture may limit their performance and generalization. Although Zhang et al. (2023) adopt a dynamic directed acyclic graph structure during inference, they merely focus on reasoning and are incompatible with external tools and diverse agents with different roles. In this work, we propose an interaction architecture that can be adjusted dynamically based on the query and be compatible with feedback and tool augmentation.

**Evaluation of the Contribution of LLM Agents**   It is non-trivial to evaluate the contribution of each LLM agent in a multi-agent system, especially when they communicate over multiple rounds. In the single-round setting, given multiple candidates to select the best one, existing methods use LLMs heavily for evaluation. However, Xiong et al. (2023) show that LLMs tend to be overconfident. For more reliable results, pairwise ranking based on an additional LLM-powered ranker has been introduced in Jiang et al. (2023) for greater accuracy. To rank $n$ responses with an independent LLM in a single round, they compare all $O(n^2)$ pairs. For better efficiency, Qin et al. (2023) use a $k$-length sliding window to choose top $k$ responses within $O(nk)$ pairwise comparisons. However, these methods have not been extended to multi-round settings. Inspired by the neuron importance score (Yu et al., 2018), we evaluate agents by propagating and aggregating single-round peer ratings. In this way, we then introduce an unsupervised metric called *Agent Importance Score* to quantify the contribution of each agent in multi-round collaborations (Section 3.3).

**Team Optimization for LLM Agents**   In terms of designing and selecting agents, Ruan et al. (2023) decompose tasks to choose or create tools accordingly. Using LLM as a planner, Lu et al.

Table 1: Comparison between **DyLAN** and representative previous works. In the second row, nodes denote agents at different time steps ($\mathbb{A}$), arrows represent edges ($\mathbb{E}$), and color indicates the role of an agent. Among these works, DyLAN is the only one which demonstrates all four key dimensions of LLM-agent collaboration, i.e., compatible with multiple roles, having an early-stopping mechanism, supporting dynamic interactions, and performing agent team optimization.

| Method | Single | LLM-Blender | PHP | Reflexion | LLM Debate | DyLAN |
|---|---|---|---|---|---|---|
| **Interaction Architecture** ($\mathbb{A}, \mathbb{E}$) | | | | | | |
| **Multiple Roles** | × | × | × | ✓ | × | ✓ |
| **Early Stopping** | × | × | × | ✓ | × | ✓ |
| **Dynamic Interactions** | × | ✓ | × | × | × | ✓ |
| **Agent Team Optimization** | × | × | × | × | × | ✓ |

(2023) sequentially select agents and tools according to their descriptions. Wang et al. (2023b) use LLMs to generate prompts for agents in response to a task query. During inference, Chen et al. (2023b) select a fixed number of agents from a set of manual prompt candidates via an additional LLM during each round of discussion. However, manual prompts require careful design, and predefined or generated descriptions may not result in the desired abilities of the agents during inference. Therefore, selecting an appropriate team of agents based on their response to a query is necessary and more appropriate. While team optimization for LLM agents is a relatively new area, human-team optimization has been studied for a long time. For instance, Liu et al. (2015) show that skill contribution is essential for selecting crowd workers to solve outsourced tasks efficiently. Based on peer rating, Lykourentzou et al. (2022) develop an algorithm for managing online workers in an optimal organization. Building upon these prior works, we introduce an automatic algorithm to optimize the team of agents by quantifying agents' contributions based on their peer ratings.

## 3 DYNAMIC LLM-AGENT NETWORK (DYLAN)

We introduce an LLM-agent collaboration framework named Dynamic LLM-Agent Network (**DyLAN**) to allow dynamic interaction between agents in LLM-agent collaborations, as illustrated in Figure 1. We formulate the systems in feed-forward networks to represent interaction architecture (Section 3.1). Under the formulation, we elaborate DyLAN's dynamic architecture designs in Section 3.2. Finally, we introduce **agent team optimization**, based on our unsupervised metric *Agent Importance Score* on top of DyLAN (Section 3.3).

### 3.1 FORMULATION

In LLM-agent collaborations, agents exchange textual messages in multi-round interactions. By viewing LLM agents at different time steps as nodes, we can treat the message exchange between multiple nodes as edges in a feed-forward network. Agents may send their reasoning steps to each other for reasoning, or send code completions for code generation, as illustrated in Appendix D.6. We then demonstrate how both the inference process and agent team optimization can be viewed as message passing algorithms over this abstract network, as an essential task-agnostic basis for DyLAN. The collaboration problem is defined to solve query $q$ by output $o$ from a given system $S$. According to the formulation, an LLM-agent collaboration comprises three components:

**Node** The node denotes an agent at a specific time step. The agent takes the contexts from other agents at the previous time step as input and generates responses based on the input query. An agent can be (I) an LLM agent that could be augmented with tools, (II) an independent tool like a script or a dedicated model, or (III) a cluster of tools. Based on the definition, we focus on textual information exchange between agents. Formally, the $i$-th agent at the $t$-th time step can be represented as a function $a_{t,i}$ mapping the agent's prompt $p_i$ (empty for (II) and (III) agents), the input query $q$, and responses from predecessor agents $\mathbb{R}_{t-1}$ to the response of itself: $r_{t,i} = a_{t,i}(p_i, q, \mathbb{R}_{t-1})$, where

$\mathbb{R}_{t-1} = \{r_{t-1,j} | j = 1, 2, ...\}$. Let $\mathbb{A}$ be the set of all nodes, $n$ be the total amount of agents, and $T$ be the maximum time step.

**Edge** Edges refer to the communication channels between nodes in $\mathbb{A}$ during the multi-agent collaboration. Let $\mathbb{E}$ represents the set of all edges in the system. The edge is directional and could be formally represented as $e = (a_{t-1,i}, a_{t,j}) \in \mathbb{E}$, where $a_{t-1,i}, a_{t,j}$ represents adjacent agents that could pass textual information: $a_{t,j}$ could perceive $a_{t-1,i}$'s output as its context. Thus, nodes are linked by edges to form the interaction architecture of agent collaboration systems, in shape of a feed-forward network $S = (\mathbb{A}, \mathbb{E})$.

**Message Passing** Message passing algorithms can guide information flow through the nodes and edges in the feed-forward network. In our LLM-agent based feed-forward network formulation, we envision two types of message passing. In a *forward passing* fashion, by passing messages to specific agents across different time steps, LLM-agent collaboration systems generate the final answer in response to the task query. Formally, we denote the output as $o$, which is the result of the query $q$ processed by system $S$ and an algorithm $f_{\text{Infer}}$ denoting the inference process explicitly (Algorithm 1 for DyLAN): $o = f_{\text{Infer}}(S, q)$. In a *backward passing* manner, The calculation process of *Agent Importance Score $\boldsymbol{I}$* is calculated by an algorithm $f_{\text{Imp}}$ to pass scores backward along edges (Algorithm 2 for DyLAN): $\boldsymbol{I} = f_{\text{Imp}}(S, q)$.

The formulation decouples the interaction architecture and the algorithm of inference process. Using this formulation, we can categorize prior works based on their interaction architectures in Table 1.

## 3.2 CONSTRUCTION OF DYLAN

With the aforementioned formulation, we step further to stack multiple layers along the temporal axis (e.g., different rounds of interaction) to build Dynamic LLM-Agent Network (**DyLAN**). To improve the efficiency and the performance of LLM-agent collaboration on various tasks, we propose to (1) add inference-time agent selection at a middle time step and (2) use Byzantine Consensus to terminate inference at a proper layer.

We define a layer as a set of agents functioning at the same time step along the temporal axis. In each layer, each node receives responses from all other nodes in the previous layer, i.e., the previous time step. Such communications form edges between layers, as shown in Figure 1. The query is fed into nodes in the first layer. Therefore, when LLM agents diverge on a query, they receive others' responses and have a debate by commenting on each other's solutions, thus deepening the reasoning procedure and reaching better consistency at a certain time step with the vantage of multiple expertise. Agents assigned diverse roles might respond from their specific perspectives.

To focus on the best solutions and reduce the computational cost, we set up the **inference-time agent selection** at $L$-th layer by selecting the top-$m$ responses to feed forward. We use an additional LLM agent, as the "LLM Ranker" in Figure 1, to analyze responses from the former layer and identify the best ones, following Jiang et al. (2023). Agents identified as not being useful are deactivated in subsequent layers, so as are the edges connecting them.

To further enhance efficiency, we introduce an **early-stopping mechanism**. Inspired by the Byzantine Consensus theory (Castro & Liskov, 1999), at least $3p + 1$ agents are needed to tolerate $p$ faulty agents in a single round of communication. Following the theory, the inference process will be terminated when over 2/3 of agents in a single layer have a consistent answer. In practice, the inference process will also be terminated when the maximum time step is reached. Note that none of the consistency measures used in prior work (Wang et al., 2023a; Aggarwal et al., 2023) applies to multi-round multi-agent interaction since their theories are assumed to execute a single LLM instance multiple times. In our case, we apply the early-stopping mechanism to each layer of DyLAN.

In short, DyLAN gives the answer in a *forward passing* manner with an efficient dynamic architecture for agent collaborations (Algorithm 1). Other implementation details are in Appendix C.1.

## 3.3 AGENT TEAM OPTIMIZATION

To automatically find the optimal team of agents among candidates for specific domains and based on their actual responses, we set up an algorithm in which the agent team optimization process could be formulated as a function $f_{\text{Optim}} : S \to S'$, where $S' = (\mathbb{A}', \mathbb{E}')$ and $\mathbb{A}'$ denotes a smaller

Table 2: Accuracy (%) on MATH dataset. CoT refers to Chain-of-Thought prompting (Wei et al., 2022) and examples are from original dataset. The number in parentheses indicates the performance difference relative to a single execution. The median of three trials is reported when non-zero `temperature` is used.

| Method | Prompting | Algebra | Counting and Probability | Geometry | Intermediate Algebra | Number Theory | Pre-Algebra | Pre-Calculus | Overall | #API Calls |
|--------|-----------|---------|--------------------------|----------|----------------------|---------------|-------------|--------------|---------|------------|
| Single Execution | CoT | 43.6 | 29.3 | 21.5 | 15.8 | 30.0 | 48.9 | 16.5 | 31.6 (+0.0) | 1.00 |
| LLM-Blender | | 47.5 | 25.5 | 23.8 | 13.8 | 39.7 | 46.7 | 15.8 | 31.7 (+0.1) | 6.00 |
| LLM Debate | | 50.2 | 25.3 | 22.3 | 13.1 | 28.9 | 48.0 | 19.0 | 32.4 (+0.8) | 8.00 |
| **DyLAN** (*Ours*) | | 52.9 | 27.2 | 25.3 | 15.5 | 33.5 | 55.2 | 19.0 | **35.7 (+4.1)** | 7.15 |
| Single Execution | Complex CoT | 49.1 | 29.7 | 22.3 | 14.6 | 33.4 | 53.8 | 16.8 | 34.1 (+0.0) | 1.00 |
| PHP | | 51.1 | 33.7 | 25.4 | 17.1 | 35.1 | 57.7 | 16.1 | 36.5 (+2.4) | 3.67 |
| **DyLAN** (*Ours*) | | 53.7 | 33.3 | 26.1 | 18.1 | 33.5 | 58.7 | 18.9 | **37.6 (+3.5)** | 6.21 |

Table 3: Accuracy (%) on MMLU dataset. "Other" stands for subjects like business, health, and misc in MMLU. The median of three trials is reported when non-zero `temperature` is used.

| Method | Humanities | Social Science | STEM | Other | Overall | #API Calls |
|--------|------------|----------------|------|-------|---------|------------|
| Random | 25.0 | 25.0 | 25.0 | 25.0 | 25.0 | - |
| Single Execution | 59.8 | 74.0 | 62.9 | 71.8 | 66.4 (+0.0) | 1.00 |
| LLM-Blender | 60.4 | 75.2 | 66.3 | 70.7 | 67.3 (+0.9) | 6.00 |
| LLM Debate | 59.8 | 77.4 | 69.0 | 75.5 | 69.3 (+2.9) | 12.00 |
| **DyLAN** (*Ours*) | 62.1 | 79.1 | 69.7 | 75.5 | **70.5 (+4.1)** | 4.39 |

but optimized composition of agents with better performance. With a pool of agent candidates, we implement $f_{\text{Optim}}$ in a three-step procedure, as shown in Figure 1: (1) *Propagation*: Take all candidates into a collaboration, ask each node to rate the solutions to the task query from its predecessors (in the forward pass); (2) *Aggregation*: Each node aggregates the ratings it has received from its successors towards itself (via another backward pass) to quantify its own contribution independently at different time steps. (3) *Selection*: During the last step, we sum up the scores for the same agent over all time steps to derive an importance score for each agent, and extract the top-$k$ agents that are most contributory according to these scores as the optimized composition.

Specifically, at the step of *Propagation*, for each agent, we ask it to rate the responses from all its predecessors. Formally, the $i$-th agent at the $t$-th time step takes its prompt $p_i$, the input query $q$, and all previous responses $\mathbb{R}_{t-1}$ from their predecessors, and further map them via a scoring function $f_{t,i}^{(s)}(\cdot, \cdot, \cdot)$ to produce the rating scores. Here, we use $w_{t-1,j,i}$ to refer to the rating score on $a_{t-1,j}$ from $a_{t,i}$, and $[w_{t-1,1,i}, w_{t-1,2,i}, ..., w_{t-1,n,i}] = f_{t,i}^{(s)}(p_i, q, \mathbb{R}_{t-1})$. After *propagation*, the contribution of node $a_{t,i}$ is the sum of its successors' contribution multiplied by their peers' ratings on the agent's response. Formally, the *aggregation* process is described as:

$$\boldsymbol{I}_{t-1,j} = \sum_{(a_{t-1,j}, a_{t,i}) \in \mathbb{E}} \boldsymbol{I}_{t,i} \cdot w_{t-1,j,i}, \tag{1}$$

where $\boldsymbol{I}_{t,i}$ denotes the contribution of $a_{t,i}$. In the final *Selection* step, *Agent Importance Score* $\boldsymbol{I}_i$ for the $i$-th agent is defined as $\boldsymbol{I}_i = \sum_{t=1}^{T} \boldsymbol{I}_{t,i}$. In practice, we initialize the contributions in the final layer first, and step backward to perform *Aggregation* layer by layer (Algorithm 2). The definition guarantees that the agent importance scores add up to 1 in each layer, which benefits fair comparison. Other details, such as initializing contributions in the final layer, are presented in Appendix C.2.

## 4 EXPERIMENTS

### 4.1 SETUP

To verify the effectiveness and efficiency of DyLAN, we conduct extensive experiments on three representative tasks including arithmetic reasoning, general reasoning, and code generation.

**Arithmetic Reasoning (AR)** We leverage MATH (Hendrycks et al., 2021b) as the evaluation dataset, which consists of 7 subareas and contains 5,000 questions in the test set. We choose LLM Debate (Du et al., 2023), LLM-Blender (Jiang et al., 2023), and the single execution on LLM as baselines. We also compared DyLAN to the previous state-of-the-art method PHP (Zheng et al., 2023). To draw a fair comparison, we categorize systems by prompting strategies, including normal CoT prompts (Wei et al., 2022) provided in the original dataset and Complex CoT provided by PHP. Since the task is sensitive to hyper-parameters, we have tuned and used each system's best `temperature`. Preliminary experiments show that collaborating agents of different roles did not introduce significant improvement, therefore we adopt agents of the same role for all systems.

**General Reasoning (GR)** For the general reasoning task, we use the MMLU dataset (Hendrycks et al., 2021a), which contains four aspects of a vast amount of problems in 57 subjects. We down-sample 1/5 of the problems in the test set because of its huge quantity. We use the same baselines as arithmetic reasoning. To align with previous works and prevent degeneration of LLM Debate, we set the hyperparameter `temperature` to 1 for all systems. The team of agents is optimized for each subject based on *Agent Importance Score* when evaluating DyLAN, as demonstrated in Table 4. The roles of candidates match the categories of MMLU, including "Mathematician" and "Programmer" for STEM, "Lawyer" and "Historian" for Humanities, "Economist" and "Psychologist" in Social Science, and "Doctor" for clinical questions in the "Other" category.

Table 4: Demonstration of experiment settings. We elaborate on the number of agents before and after optimization and tool usage in DyLAN of each task.

| Task | Agent Team Size | Tool Usage |
|------|-----------------|------------|
| AR | 4 | × |
| GR | $7 \to 4$ | × |
| CG | $12 \to 8$ | ✓ |

**Code Generation (CG)** We use the HumanEval benchmark in the code generation task, with 164 human-labeled function-level completion codes and unit tests (Chen et al., 2021). We found that LLM Debate and LLM-Blender could not feasibly adapt to this task in their implementation, thus they are excluded them from the baselines. As a result, we leverage two strong methods CodeT (Chen et al., 2023a) and Reflexion (Shinn et al., 2023) along with the single execution as the baselines. The team of agents is optimized in DyLAN according to Table 4. Additionally, we require 4 of the 8 agents should be the role of code writer and the other 4 are judges providing code reviews in different time steps. And two of the judges are individually augmented with a code interpreter and a syntax checker. Please refer to Appendix C.1 for more details.

## 4.2 MAIN RESULTS

In Table 2, Table 3, and Table 5, we report the classification accuracy on MATH and MMLU, and Pass@1 on HumanEval, respectively. Additionally, we recorded the average times of calling LLMs of each method on each query as #API calls. This metric serves as a proxy for the computational cost and the efficiency of architectures for LLM-agent collaborations, which cannot be determined from the number of tokens since the number varies greatly depending on the query.

**DyLAN improves overall performance in different tasks.** Table 2 shows that DyLAN has significantly higher performance than baselines on arithmetic reasoning, demonstrating the advantage of the dynamic architecture. Relative to a single execution, it gains a +4.1 improvement with CoT prompts and +3.5 with Complex CoT prompts, demonstrating robustness against different prompt-

Table 5: Experimental results on HumanEval dataset. We indicate the foundation model of methods except for GPT-35-turbo. The median of three trials is reported when non-zero `temperature` is used.

| Method | Pass@1 | #API Calls |
|--------|--------|------------|
| Single Execution | 73.2 (+0.0) | 1.00 |
| CodeT | 65.8 (-7.4) | 20.00 |
| CodeT (Codex) | 74.8 (+1.6) | 20.00 |
| Reflexion | 68.3 (-4.9) | 4.05 |
| **DyLAN** (*Ours*) | **82.9 (+9.7)** | 16.85 |
| Single Execution (GPT-4) | 88.4 (+15.2) | 1.00 |
| Reflexion (GPT-4) | 91.4 (+18.2) | 7.32 |
| **DyLAN** (*Ours*, GPT-4) | **92.1 (+18.9)** | 15.94 |

ing strategies. In Table 3 and Table 5, DyLAN also significantly improves the performance by +4.1 and +9.7 on general reasoning and code generation tasks, respectively, relative to the single execution. We argue part of the improvement can be attributed to the dynamic multi-path architecture, which allows different opinions to be delivered simultaneously. In contrast, for methods in sequential architecture like PHP (Zheng et al., 2023), incorrect intermediate answers might easily influence

the final output due to only one reasoning path. It is the same for Reflexion (Shinn et al., 2023) in code generation, in which false tests may mislead the self-debugging process from our observations. In our case, any feedback from predecessors could be rated by the nodes at any time step, making it easier to identify false intermediate feedback explicitly.

**DyLAN has a reasonable computational cost.** From Table 2, we find DyLAN realizes an 10.2% improvement to LLM Debate in terms of accuracy, with 10.6% lower #API calls (L3 vs. L4), suggesting it is a better trade-off between efficiency and effectiveness. Similar trends can be observed in Table 3. DyLAN has a better overall accuracy with only 36.6% API calls of LLM Debate (4.39 vs. 12.00). Moreover, we see that DyLAN dynamically adjusts the cost based on the difficulty of a query. Given that questions in MMLU are less challenging than in MATH, DyLAN has 2.76 fewer #API calls on the query from MMLU compared to one from MATH. This makes sense as generally, the more complex the task, the harder it will be to come to a consensus.

**DyLAN benefits from agent team optimization.** Additionally, we found that an optimized team of agents could enhance DyLAN. For different subjects in MMLU, agent compositions are adjusted correspondingly to improve up to 25.0% in accuracy, as shown in Table 6. After agent team optimization, we find a smaller set of agents are selected from the given large pool of candidates for specific subjects, resulting in significant performance improvement, such as "Mathematician", "Programmer", and "Economist" for college mathematics, and "Doctor" and "Psychologist" for clinical knowledge. Moreover, before

Table 6: The optimized composition of roles and performance improvement in different subjects of MMLU dataset when optimizing DyLAN of 7 agents to 4 agents.

| Subject | Optimized Composition | Performance Improvement |
|---|---|---|
| college mathematics | Economist, Lawyer, Programmer, Mathematician | **25.0** : 40.0 → 65.0 |
| management | Lawyer, Psychologist, Economist, Programmer | **14.3** : 76.2 → 90.5 |
| high school statistics | Historian, Programmer, Psychologist, Mathematician | **9.3** : 65.1 → 74.4 |
| clinical knowledge | Doctor, Mathematician, Programmer, Psychologist | **5.7** : 69.8 → 75.5 |
| public relations | Historian, Psychologist, Lawyer, Mathematician | **4.5** : 54.5 → 59.1 |

performing agent team optimization, i.e., all the 12 agents are used, the performance of DyLAN on HumanEval is 76.2, which means agent team optimization introduces an improvement of +6.7 points (76.2 → 82.9), suggesting that *Agent Importance Scores* can effectively capture and reflect the actual abilities of agents to particular task domains.

## 4.3 Ablation Studies

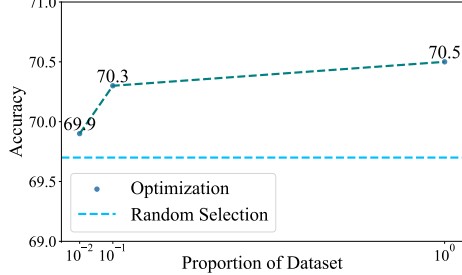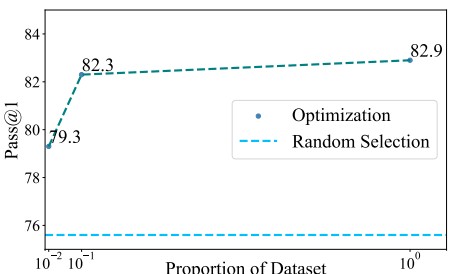

Figure 2: Experimental results of optimizing agent team with small subsets of MMLU (left) and HumanEval (right). X-axis denotes the proportion to the original datasets. We also randomly selected the same amount of agents as a baseline.

**Data Efficiency of Agent Team Optimization** We further demonstrate the data efficiency of agent team optimization by performing it based on different amount of data. The experiments are conducted on the MMLU and HumanEval datasets. We sample the subsets with the proportions of 0.01 and 0.1 of the original dataset. Agent team optimization is performed on the subsets and tested the optimized team of agents on the whole dataset. The settings are identical to experiments in Section 4.1. As shown in Figure 2, with the optimized team of agents on 0.1 of the original dataset, DyLAN has demonstrated similar performance compared to optimizing on the whole dataset, with

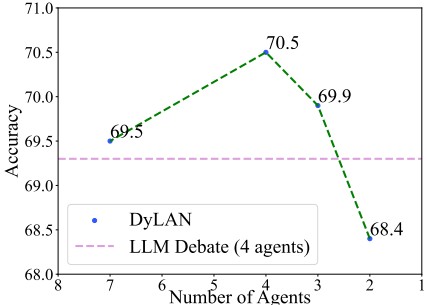 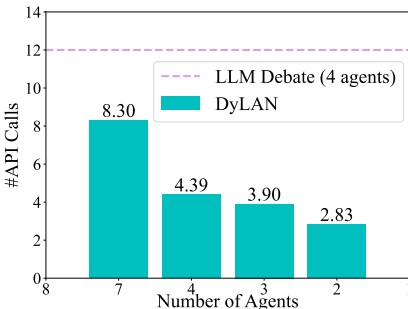

Figure 3: Impact of optimized agent team size. 2∼4 agents are selected from 7 candidate agents based on *Agent Importance Score*. Accuracy (left) and #API calls (right) on MMLU are reported.

only 0.2 loss of overall accuracy on MMLU and 0.6 loss of Pass@1 on HumanEval. Especially, as code review has significant impact on the performance on code generation, it is critical to select a proper set of judges to provide code review. We can observe that even with only 0.01 of the original dataset, DyLAN could obtain a significant improvement of +3.7 over random selection on HumanEval, indicating its effectiveness on selecting proper team of agents.

**Impact of Optimized Agent Team Size** As shown in Figure 3, DyLAN with an optimized team of 3 agents can outperform both the same architecture with 7 agents and LLM Debate with 4 agents, suggesting the effectiveness of our proposed agent optimization. The efficiency is also significantly improved by 52.9% and 67.8%, respectively. These results indicate that an optimized selection of agents would be able to collaborate on tasks better and reach the consensus much faster and more accurately than many agents thrown together at random.

**Impact of Early-Stopping and Inference-Time Agent Selection** As shown in Table 7, early-stopping mechanism boosts efficiency to a great extent by minimizing #API calls by 45.0%, 66.2%, and 11.3% on MATH, MMLU, and HumanEval respectively, while providing slight performance improvement. Inference-time agent selection, however, is critical to enhance the correctness of the final answer. We conjecture it is because low-performing agents are filtered at specific time steps.

**Stability of DyLAN with Different Backbone Models** There is also a notable difference in code generation tasks when the backbone model changes (Table 5). Reflexion and CodeT's performances are heavily related to the backbone model (L4 vs. L5 and L6 vs. L9). Instead, DyLAN shows a steady, consistent high performance (L7 vs. L10) under different backbone models with almost the same amount of API calls.

Table 7: Ablation study on the early-stopping mechanism (*es*) and the inference-time agent selection component (*its*). #API denotes the average number of API calls.

| Method | MATH | | MMLU | | HumanEval | |
|---|---|---|---|---|---|---|
| | Acc. | #API | Acc. | #API | Pass@1 | #API |
| **DyLAN** | **35.7** | **7.15** | **70.5** | **4.39** | **82.9** | **16.85** |
| *w/o es* | 35.0 | 13.00 | 70.1 | 13.00 | 80.5 | 19.00 |
| *w/o its* | 33.8 | 8.20 | 69.9 | 7.05 | 76.2 | 17.98 |

## 5 CONCLUSION AND FUTURE WORK

This work introduces a framework named Dynamic LLM-Agent Network (DyLAN) for LLM-agent collaboration on complicated tasks. DyLAN enables agents to interact for multiple rounds in a dynamic architecture with inference-time agent selection and an early-stopping mechanism to improve performance and efficiency. We further design an automatic agent team optimization algorithm based on an unsupervised metric termed *Agent Importance Score*, enabling the selection of best agents based on the contribution each agent makes. Overall, DyLAN reveals significant improvement on diverse tasks with relatively less computational cost compared to baselines.

In this work, the agents are driven by GPT-3.5-turbo and GPT-4, we will explore the effectiveness of DyLAN on agents built on open-source foundation models in future. And it is also worth to extend DyLAN to more complicate scenarios such as software development, virtual room chat, video games, and so on (Hong et al., 2023; Nascimento et al., 2023; Zhou et al., 2023; Zhu et al., 2023; Chan et al., 2023).

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

## A  BROADER IMPACT

LLM-agent systems are widely used in practical applications. DyLAN could also effortlessly cover practical software development, virtual room chat, video games, and so on (Hong et al., 2023; Nascimento et al., 2023; Zhou et al., 2023; Zhu et al., 2023; Chan et al., 2023). In these open-world environments, agents may operate as planners, actors, etc. DyLAN only requires people to give rough instructions on the constitute of agents and could automatically optimize a better team of agents to construct an efficient multi-agent system. These systems could benefit from DyLAN to reduce human labor on designing agents and have a better performance on their target tasks.

Also, the overall architecture of DyLAN (Figure 1) reflects the optimal collaboration organization of human online workers (Lykourentzou et al., 2022), and reveals significant performance in agent collaborations. Therefore, simulating human collaboration by LLM-agent collaborations under DyLAN might also be possible. Optimizing human collaboration by searching and simulating LLM agents will hopefully be more convenient and effective.

## B  DISCUSSION & LIMITATION

In experiments, we view code generation tasks as representative of open-ended generation tasks and adopt `BLEU` to decide whether two answers are consistent in Byzantine Consensus. In fact, the performance could be further leveraged by task-specific methods like CodeBLEU (Ren et al., 2020) or CodeT (Chen et al., 2023a).

For practical usage, the agent-evaluation metrics could cooperate with human annotation to give a more precise evaluation result, mainly when facing data scarcity problems. We apply agent team optimization on DyLAN simply on top of the inference-time agent selection. It still remains to be seen how to cooperate offline and online optimization methods in a finer granularity to further improve performance and efficiency in LLM-agent collaboration systems.

Additionally, in imbalanced cases where the majority of agents are designed to contradict the task requirement, low performance might be caused, though agent team optimization could differentiate top contributory agents. To tackle the imbalance of high- and low-performing agents, replicating agents with high *Agent Importance Score* instead of including low-score agents could be a solution. Additionally, in extreme circumstances, we might need to automatically introduce agents from more capable LLMs with validation, in addition to agent team optimization.

## C  IMPLEMENTATION DETAILS

### C.1  DETAILED EXPERIMENT SETTINGS

**Common Settings**  In all experiments, we use `gpt-35-tubo-0301` for every LLM agent if not specified. In Table 5, "(GPT-4)" denotes `gpt-4-0613` and "(Codex)" denotes `code-davinci-002` from OpenAI (Chen et al., 2021; OpenAI, 2023). To avoid the context length issue in prior work (Du et al., 2023; Liu et al., 2023a), we set memory space for agents in DyLAN to 1 only to keep the freshest responses of predecessors. We set `max tokens` to 2048

**Algorithm 1:** The Inference Process $f_{\text{Infer}}$ of **Dy-LAN** on an arbitrary query

---

**Data:** Query $q$, DyLAN $S = (\mathbb{A}, \mathbb{E})$, task-specific answer extraction method ans

**Result:** Output $o$

/* $\mathbb{E} = \{(a_{t,i}, a_{t+1,j})\}_{t=1}^{T-1}, a_{t,i}, a_{t+1,j} \in \mathbb{A}$ */

**for** $t = 1; T$ **do**

  **if** $t = L$ **then**

    /* inference-time agent selection */

    $\mathbb{R}_{top} \leftarrow \text{top} - m(\{r_{t-1,j} | a_{t-1,j} \in \mathbb{A}\})$;

    $\mathbb{E} \leftarrow \mathbb{E} \backslash \{(a_{t',j}, *), (*, a_{t',j}) | r_{t',j} \in \mathbb{R}_{top}, t' \geq t - 1\}$;

    $r_{t,j} \leftarrow r_{t-1,j}, \forall (a_{t-1,j}, a_{t,j}) \in \mathbb{E}$;

  **else**

    $r_{t,i} \leftarrow a_{t,i}(p_i, q, \{r_{t-1,j} | (a_{t-1,j}, a_{t,i}) \in \mathbb{E}\}), \forall i, \exists k, (a_{t,i}, a_{t+1,k}) \in \mathbb{E}$;

  **end**

  **if** *Byzantine Consensus is reached* **then**

    break;

  **end**

**end**

/* extract final answer */

$o \leftarrow \text{ans}(\{r_{t,i} | a_{t,i} \in \mathbb{A}\}, q)$;

---

**Algorithm 2:** The Calculation Process $f_{\text{Imp}}$ of *Agent Importance Score* within **DyLAN**

---

**Data:** Output $o$, DyLAN $S = (\mathbb{A}, \mathbb{E})$

**Result:** *Agent Importance Score* of agents $\boldsymbol{I}$

flag $\leftarrow$ False;

**for** $t = T; 1$ **do**

  **if** $\{a_{t,i} | \exists k, (a_{t-1,k}, a_{t,i}) \in \mathbb{E}\} \neq \phi$ **then**

    **if** $\neg$flag **then**

      flag $\leftarrow$ True;

      /* Initialzation */

      distribute scores for $I_{t,i}$;

    **else**

      $\mathbb{R}_{t-1} \leftarrow \{r_{t-1,j} | (a_{t-1,j}, a_{t,i}) \in \mathbb{E}\}$;

      $[w_{t-1,1,i}, ..., w_{t-1,m,i}] \leftarrow a_{t,i}^{-1}(p_i, q, \mathbb{R}_{t-1})$;

      $\boldsymbol{I}_{t-1,j} \leftarrow \boldsymbol{I}_{t-1,j} + \boldsymbol{I}_{t,i} w_{t-1,j,i}, a_{t-1,j} \in \{a_{t-1,j} | (a_{t-1,j}, a_{t,i}) \in \mathbb{E}\}$;

    **end**

  **end**

**end**

---

to avoid exceeding the maximum context length. We use a listwise ranker in the inference-time agent selection of DyLAN because of the effectiveness and efficiency, compared to ELo rating (Herbrich et al., 2006) or Sliding Window (Qin et al., 2023) we have tested in Appendix D.3. We use the same ranker to implement LLM-Blender (Jiang et al., 2023) in experiments. We set $m = 2$ in the inference-time agent selection, because it's the minimal number for collaborations and we empirically found it brings great trade-off between effectiveness and efficiency. To avoid positional bias, we shuffle the responses from agents at $(L-1)$-th time step before performing inference-time agent selection. The detailed algorithm is in Algorithm 1. To implement the early-stopping mechanism, we need to determine whether the answers from the nodes in the same layer of DyLAN are consistent. For classification problems, the answers are consistent if identical, and for open-ended generation, the consistency is determined by a threshold of `BLEU` score.

**Experiments on Reasoning Tasks** In general reasoning, we extract the answer from the response by matching the last "(X" or "(X)", where "X" represents A, B, C or D. On average, DyLAN with the 7 agents consumes 8.30 API calls during the agent team optimization. Inference-time agent selection functions on the third round ($L = 3$ in Algorithm 1). They could go through at maximum $T = 4$ rounds of interaction. We also searched `temperature` in $\{0, 0.2, 0.8, 1.0\}$ for the best configuration for each system. In arithmetic reasoning, we set `temperature` to 0 for the single execution and PHP, 0.2 for LLM Debate, LLM-Blender, and DyLAN with Complex CoT prompts, and 1.0 for DyLAN with simple CoT prompts in Table 2, since systems with the same prompts will give all the same responses if `temperature` is zero, causing degradation. Also, we did not optimize the team of agents for DyLAN on arithmetic reasoning tasks. We follow the answer extraction method from the origin paper (Hendrycks et al., 2021b). We construct DyLAN with 4 agents assigned no specific roles and let agents to interact for at maximum 4 rounds. We reported the classification accuracy of each category averaged across subjects and the numbers of API calls of running DyLAN on the optimized team of agents. In DyLAN, inference-time agent selection functions at the third time step ($L = 3$). Furthermore, we use the exact match to determine the consistency of answers in the early-stopping mechanism and extract the final answer from nodes in the last layer on reasoning tasks.

**Experiments on Code Generation Tasks** In the code generation task, we set `temperature` to 0 for the single execution, Reflexion, and DyLAN, and 0.8 for LLM Debate, LLM-Blender, CodeT, and DyLAN in Table 5. In DyLAN, we optimized four agents to write code and four agents to give

feedback from the last 12 candidate roles in Appendix E. The optimization is also conducted on `gpt-35-tubo-0301`. The selected code writers are "Python Assistant", "Algorithm Developer", "Computer Scientist", and "Programmer"; and the selected judges are "Syntax Checker", "Unit Tester", "Reflector", and "Ranker". "Syntax Checker" is pure external tools without LLMs and "Unit Tester" is equipped with a code interpreter. We reported the numbers of API calls of running DyLAN on the optimized team of agents. On average, during the agent team optimization, DyLAN with the 12 agents consumes 23.04 API calls. In DyLAN, solutions given by code writers are reviewed by judges in at maximum $T = 6$ rounds. At time step $t = 1, 3, 4, 6$, code writers gives solutions and judges review it at $t = 2, 5$ ($L = 4$). Specifically, early-stopping mechanism functions at the third layer and later ($t \geq 3$). We use BLEU score in the early-stopping mechanism. We calculate BLEU by *sacreBLEU*[2] (Post, 2018). For answer extraction, we store all unit tests from the unit tester (if exists in the system) and randomly select the final output from the top 5 code completions from all nodes that pass most tests.

## C.2 CALCULATION OF AGENT IMPORTANCE SCORE

To implement the agent team optimization algorithm under DyLAN, only one sentence needs to be injected into the end of the prompt of each node in DyLAN: `Along with the answer, give a score ranging from 1 to 5 to the solutions of other agents. Put all {`$num_{\text{predecessor}}$`} scores in the form like [[1, 5, 2, ...]]`, where $num_{\text{predecessor}}$ denotes the number of predecessors of the node. The prompt functions as the $f_{t,i}^{(s)}$ in Section 3.1 and we extract $w_{t,j,i}$ from its response at the same time when we extract the message that passes between nodes. To avoid positional bias, we shuffle responses from agents at previous time step when rating.

In Algorithm 2, initial contributions are distributed on nodes at the last layer. For reasoning tasks, we uniformly distribute contributions to agents that give consistent answers in the last layer. On code generation tasks, we uniformly distribute contributions in the final round with no syntax error in their answers. During agent team optimization, we independently optimize the composition of agents for each domain. Thus, in experiments on general reasoning and code generation tasks, we sum up agent importance scores to optimize a team of agents for each subject on the MMLU dataset and for the HumanEval dataset, respectively.

## D ADDITIONAL RESULTS

In this section, detailed results and additional experiments are presented.

### D.1 HUMAN PRIORS AND AUTOMATIC EVALUATION RESULTS

We further investigated how these agents selected by our unsupervised metric *Agent Importance Score* differ from human priors (e.g., these predefined roles). To do so, we calculated *Agent Importance Scores* for 7 agents on each subject of the MMLU

Table 8: Subjects on which agents have the top-ranked *Agent Importance Score* in the experiment with DyLAN of 7 agents on MMLU dataset. Green annotation denotes the fields related to the role from the human perspective, which are annotated manually.

| Role | Doctor | Programmer |
|---|---|---|
| **Top 10 Subjects** | high school computer science
clinical knowledge
college biology
professional medicine
nutrition
high school US history
human aging
anatomy
high school biology
high school psychology | high school physics
electrical engineering
high school government and politics
college computer science
college chemistry
high school mathematics
formal logic
abstract algebra
machine learning
computer security |

dataset. As an example, we show the subjects where the agent of "Doctor" and "Programmer" has the highest *agent importance score* among all agents in Table 8 and Table 9. Though most subjects seems to be reasonably aligned with the role of the agent based on human priors (with green annotations), there are some subjects that do not match human priors, e.g., *high school computer science* as the subject that "Doctor" has the highest score. It exhibits the difference between human priors and the evaluation results *Agent Importance Scores* on other agents.

We also compare current agent team optimization method that is implemented with *Agent Importance Score* with the implementation with *Human Prior Selection* on a few subjects in MMLU and HumanEval. For *Human Prior Selection*, we setup an LLM to select the agents for collaborations

---

[2]The signature of *sacreBLEU* is "nrefs:1|case:mixed|eff:no|tok:13a|smooth:exp|version:2.3.1".

Table 9: Subjects on which agents have the top-ranked *Agent Importance Score* in the same experiment in Table 8. Green annotation denotes the fields highly related to the role from the human perspective.

| Role | Mathematician | Lawyer | Historian | Economist | Psychologist |
|---|---|---|---|---|---|
| **Top 10 Subjects** | college physics
US foreign policy
college computer science
econometrics
marketing
high school mathematics
abstract algebra
international law
professional accounting
human sexuality | high school microeconomics
medical genetics
prehistory
sociology
human aging
management
formal logic
world religions
jurisprudence
international law | US foreign policy
econometrics
world religions
public relations
high school government and politics
philosophy
astronomy
high school statistics
machine learning
high school European history | high school computer science
jurisprudence
logical fallacies
professional accounting
high school microeconomics
high school European history
computer security
moral disputes
professional law
college mathematics | global facts
public relations
business ethics
high school US history
philosophy
moral disputes
management |

based on the description of the task and role prompts of each agent (prompt templates is demonstrated in Appendix E). As shown in Table 10, the implementation with *Agent Importance Score* steadily outperforms Human Prior Selection. There are two major reasons: (1) Compared to posterior optimization methods, prior selection may not grasp the actual behaviors of agents, and may not understand which agents are most contributory or helpful to others in the real collaboration process. Thus, in High School Statistics, Clinical Knowledge, and Public Relations subjects of MMLU dataset, prior selection performs even worse than random selection. (2) *Human Prior Selection* might struggle to understand tool augmentation without peer ratings from fellow agents. From our observation, Unit Tester and Syntax Checker were not selected for code generation tasks, which may cause lower performance.

Table 10: Performance of different implementations of agent team optimization on five subjects of MMLU (top) and HumanEval (bottom). The five subjects of MMLU and other settings are identical to Table 6. The overall accuracy in the left table denotes the accuracy across the five subjects.

| Agent Team Size | Optimization Method | College Mathematics | Management | High School Statistics | Clinical Knowledge | Public Relations | Overall |
|---|---|---|---|---|---|---|---|
| 7 | *(before optimization)* | 40.0 | 76.2 | 65.1 | 69.8 | 54.5 | 63.5 (+0.0) |
| 4 | *Random Selection*
*Human Prior Selection*
*Agent Importance Score* | 45.0
60.0
**65.0** | 71.4
80.1
**90.5** | 67.4
65.1
**74.4** | 71.7
69.8
**75.5** | 54.5
54.5
**59.1** | 64.8 (+1.3)
66.7 (+3.2)
**73.6 (+10.1)** |

| Agent Team Size | Optimization Method | Pass@1 | #API Calls |
|---|---|---|---|
| 12 | *(before optimization)* | 76.2 (+0.0) | 23.04 |
| 8 | *Random Selection*
*Human Prior Selection*
*Agent Importance Score* | 75.6 (-0.6)
78.0 (+1.8)
**82.9 (+6.7)** | 17.73
16.37
16.85 |

## D.2  STABILITY OF DYLAN ON TEMPERATURE

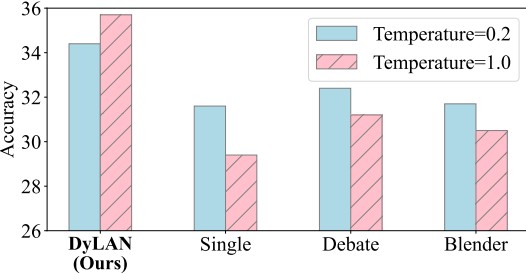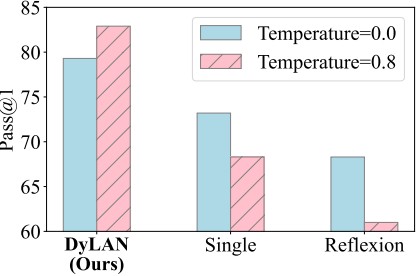

Figure 4: Performance of different methods under low and high temperatures on MATH (left) and HumanEval (right) datasets. DyLAN shows better robustness to different temperature and even takes advantage of higher temperature.

We tested a few methods on MATH (with simple CoT prompts) and HumanEval datasets under both low and high temperatures and repeated each experiment 3 times when the temperature was not 0.

We exhibit the experimental results in Figure 4. From experimental results, we found that DyLAN is more stable on different hyper-parameters.

Experiments show that `temperature` greatly influences arithmetic reasoning and code generation tasks. In Figure 4, we found that most baseline methods have significant performance drops when temperature increases, but DyLAN shows strong robustness to various temperatures. We surprisingly found that DyLAN gets better results when temperature rises, suggesting it has benefited from diversity instead of being disturbed by low-quality answers of high-temperature agents. The inference-time agent selection may lead to the higher accuracy by keeping best responses when agents' replies become more diverse. In conclusion, the collaboration of different roles functions effectively and robustly in the dynamic architecture. Nonetheless, higher temperature requires DyLAN to take more API calls (about +0.98 on average on MATH (`temperature`: $0.2 \rightarrow 1.0$)).

### D.3 DIFFERENT RANKING METHODS

We also tested different ranking methods for inference-time agent selection of DyLAN on the MMLU dataset. We tested listwise ranker with our own prompts, pairwise GPT ranker from original LLM-Blender (Jiang et al., 2023), Elo Score from TrueSkill (Herbrich et al., 2006) also implemented with pairwise ranker, and pairwise ranker with Sliding Window algorithm (Qin et al., 2023). In Table 11, we show that different ranking methods have a relatively low impact on performance, probably because of strong discrimination ability of `GPT-3.5`, but pairwise ranking methods al-

Table 11: Overall accuracy (%) of DyLAN with different ranking method in the inference-time agent selection on MMLU dataset. Other settings are identical with Table 3.

| Ranking Method | | Overall Accuracy | #API Calls |
|---|---|---|---|
| **Listwise Ranker** | | **70.5** | **4.39** |
| Pairwise | LLM-Blender | 70.1 | 19.27 |
| | Elo Score | 70.3 | 19.55 |
| | Sliding Window | 70.3 | 11.40 |

ways consume higher computational cost. Thus, we chose a listwise ranker in our implementation of DyLAN.

### D.4 RELATION BETWEEN HUMAN PRIORS AND OPTIMIZED TEAMS OF AGENTS

In the agent team optimization experiments on the DyLAN of 7 agents with different roles, we recorded the in-domain rate with different numbers of remaining roles in Table 12. The in-domain rate is the proportion of teams with at least one agent matching the domain of the subject based on human priors (e.g. "Programmer" matches STEM subjects, as described in Section 4.1), given that the teams are optimized on each subject. The results suggest that the optimized team of agents is roughly aligned with human priors but still with a significant offset.

Table 12: In-domain rate of teams that contain roles in the domain of the subject after optimization on the DyLAN of 7 agents with distinct roles.

| #Roles | 7 | 6 | 5 | 4 | 3 | 2 | 1 |
|---|---|---|---|---|---|---|---|
| **In-Domain Rate** | 1.00 | 0.97 | 0.91 | 0.88 | 0.74 | 0.60 | 0.39 |

### D.5 SHAPLEY VALUE V.S. AGENT IMPORTANCE SCORE

*Shapley Value* is a widely used supervised metric for evaluating contribution of a single agent in a multi-agent system. Though it is not suitable for unsupervised agent team optimization, we use it for validating *Agent Importance Score*. We implement a simplified algorithm for LLM-agent collaboration systems. Given that the collaboration process is symmetric to roles in the formulation of the feed-forward network, we could reduce the permutation set in the original formula (Lundberg & Lee, 2017) to the combination set:

$$S_i(\mathbb{R}) = \frac{1}{|\mathbb{C}||\mathbb{R}|} \sum_{\mathbb{T} \in \mathbb{C}} (\text{Performance}(\mathbb{T} \cup \{i\}) - \text{Performance}(\mathbb{T})), \tag{2}$$

where $\mathbb{R}$ is the set of agents in the system, $\mathbb{C}$ is the combination set of $\mathbb{R}\backslash\{i\}, i \in \mathbb{R}$, and Performance denotes the overall performance of the system on the current task, e.g., classification accuracy or Pass@1. The metric requires ground truth and multi-pass results of the system with different subsets of agents. We use classification accuracy for classification tasks and Pass@1 for code generation tasks. However, its computation cost is still too high when the number of agents grows larger due to its combinatorial complexity.

To examine *Shapley Value* as an indicator of agent team optimization, we also randomly chose three combinations of three roles out of all 7 roles to assemble DyLAN with three agents and calculated the *Shapley Value* and the *Agent Importance Score* in the 3-agent DyLAN on MMLU. In Table 13, we report the correlations between *Shapley Values* and *Agent Importance Scores* on the 3-agent experiments. We are curious whether *Agent Importance Score* is an unsupervised substitution for *Shapley Value*. So, we calculated the KL divergence and a listwise loss (ListMLE (Xia et al., 2008)) between *Agent Importance Scores* and *Shapley Value*. It indeed shows a high correlation between the distributions of the two metrics when at least one agent is in the domain of the question (in column **In-Domain**).

In summary, while *Shapley Value* is a self-evident metric for individual contribution, *Agent Importance Score* emerges as a promising, unsupervised alternative with light computational complexity.

Table 13: Correlation between different metrics for quantifying agents' contributions in 3-agent DyLAN on MMLU dataset. We compute the KL divergence $D_{\mathrm{KL}}$ and the ListMLE loss $\mathcal{L}_{\mathrm{ListMLE}}$ between Shapley Value and other metrics on each subject and report the average value. The **In-Domain** column means at least one agent in DyLAN matches the subject according to Appendix C.1, and **Off-Domain** means none of agents matches the subject.

| Metric | In-Domain | | Off-Domain | |
|---|---|---|---|---|
| | $D_{\mathrm{KL}}$ | $\mathcal{L}_{\mathrm{ListMLE}}$ | $D_{\mathrm{KL}}$ | $\mathcal{L}_{\mathrm{ListMLE}}$ |
| Shapley Value | 0 | 0.673 | 0 | 0.674 |
| Agent Importance Score | $\mathbf{0.229 \times 10^{-3}}$ | **0.686** | $0.347 \times 10^{-3}$ | 0.693 |
| Uniform Distribution | $0.359 \times 10^{-3}$ | 0.693 | $0.327 \times 10^{-3}$ | 0.693 |

## D.6 CASE STUDY

In Figure 5 and Figure 6, we demonstrate the cases of DyLAN on the code generation and the general reasoning tasks, respectively. First, we notice that the interaction architecture is different between figures, exhibiting dynamic architecture of DyLAN on different queries. The former gives answer at $t = 4$, the latter at $t = 2$. We also notice that the answer is gradually growing better along the temporal axis. In Figure 6, only the agent with "Mathematician" role gives a correct answer and then convince other agents, suggesting that proper agents are critical to task performance and our agent team optimization method is effective. Although there are hallucinations when agents respond to the query or evaluate predecessors' responses, the overall solution is correct, and the distribution of *Agent Importance Score* is reasonable. Also, instructing LLM agents to rate scores on predecessors hints them to reflect on predecessors' responses, which might be helpful to give better answers. Last but not least, agents with different roles lead a diverse conversation and make full use of it, which benefits performance and robustness.

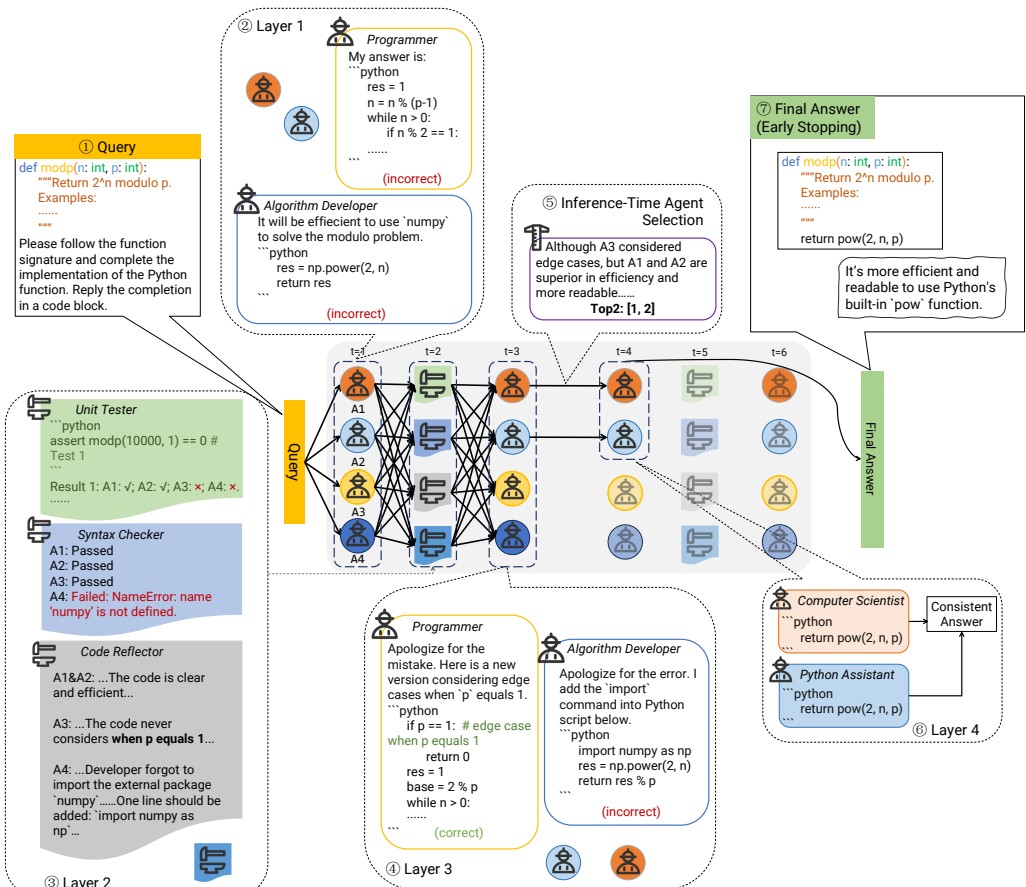

Figure 5: A case of DyLAN solving code generation task. Different agents are recruited to write code and give feedback. At the time steps $t = 2, 5$, we ask judges to provide code reviews. The result grows better layer by layer regarding correctness, efficiency, and readability. Different directions of implementation are delivered forward in implicit multiple paths. We ignore the peer rating scores in responses of agents for computing *Agent Importance Scores*.

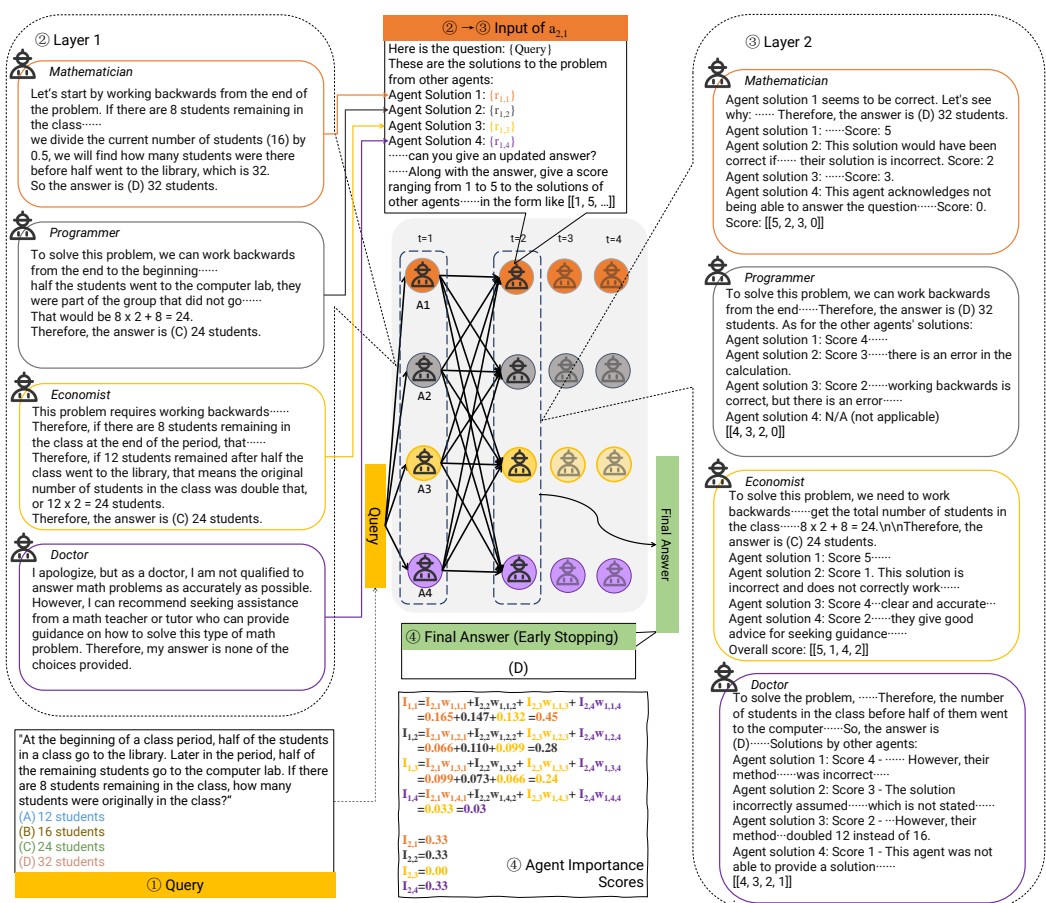

Figure 6: A case of DyLAN solving general reasoning task. Different agents are recruited to give and refine solutions. The result is incorrect at the first time step but correct at the second time step. It includes the ratings from agents for calculating *Agent Importance Scores*.

# E  PROMPTING TEMPLATES

In DyLAN, agents are assigned roles extracted from an open-source code base[3], relative research projects (Du et al., 2023; Shinn et al., 2023; Zheng et al., 2023), and GPT-4 generation with minor modification. We exhibit the instruction templates of different datasets and the prompts of all roles in Table 14.

| Role | Prompt |
|---|---|
| MMLU Instruction | Here is the question: {question}\n\nThese are the solutions to the problem from other agents: {responses}\n\nUsing the reasoning from other agents as additional advice with critical thinking, can you give an updated answer? Examine your solution and that other agents step by step. Notice that their answers might be all wrong. Put your answer in the form (X) at the end of your response. (X) represents choice (A), (B), (C), or (D). |
| MATH Instruction | Follow the given examples and answer the mathematics problem.\n\n{question}\n\nThese are the solutions to the problem from other agents: {responses}\n\nUsing the reasoning from other agents as additional advice with critical thinking, can you give an updated answer? Examine your solution and that other agents step by step. Notice that their answers might be all wrong. |
| HumanEval Instruction | You must complete the python function I give you by rectifying previous implementations. Use the other information as a hint.\nBe sure to use the same indentation I specified. Furthermore, you may only write your response in code/comments.\n[improved impl]:\n```python\n{function signature}\n```\n \nPlease follow the template by repeating the function signature and complete the new implementation in [improved impl]. If no changes are needed, simply rewrite the implementation in the Python code block. |
| Human Prior Selection Instruction | A few agents will collaborate on the same task query. Please select the optimal composition of the candidate agents based on the description of the task and the agents' profiles.\n\n Task: {subject}\n- Agents\n{ - Agent/Code Writer/Judge t (Name): Role Prompt/Description\n}$_{t=1}^{n}$\n\nWe want to select k agents/code writers/judges among these candidates. Please write the agent IDs as the following format: [1, 2, 3, 4]. There could be multiple agents with the same ID. |
| Mathematician | You are a mathematician. You are good at math games, arithmetic calculation, and long-term planning. |
| Programmer (MMLU) | You are a programmer. You are good at computer science, engineering, and physics. You have experience in designing and developing computer software and hardware. |
| Lawyer | You are a lawyer. You are good at law, politics, and history. |
| Historian | You are a historian. You research and analyze cultural, economic, political, and social events in the past, collect data from primary sources and use it to develop theories about what happened during various periods of history. |
| Economist | You are an economist. You are good at economics, finance, and business. You have experience on understanding charts while interpreting the macroeconomic environment prevailing across world economies. |
| Psychologist | You are a psychologist. You are good at psychology, sociology, and philosophy. You give people scientific suggestions that will make them feel better. |

---

[3]https://github.com/GoGPTAI/ChatGPT-Prompt/blob/main/prompts.csv

| Role | Prompt |
|---|---|
| Doctor | You are a doctor and come up with creative treatments for illnesses or diseases. You are able to recommend conventional medicines, herbal remedies and other natural alternatives. You also consider the patient's age, lifestyle and medical history when providing your recommendations. |
| Python Assistant | You are a Python writing assistant, an AI that only responds with python code, NOT ENGLISH. You will be given a function signature and its docstring by the user. Write your full implementation (restate the function signature). |
| Algorithm Developer | You are an algorithm developer. You are good at developing and utilizing algorithms to solve problems. You must respond with python code, no free-flowing text (unless in a comment). You will be given a function signature and its docstring by the user. Write your full implementation following the format (restate the function signature). |
| Computer Scientist | You are a computer scientist. You are good at writing high performance code and recognizing corner cases while solve real problems. You must respond with python code, no free-flowing text (unless in a comment). You will be given a function signature and its docstring by the user. Write your full implementation following the format (restate the function signature). |
| Programmer (HumanEval) | You are an intelligent programmer. You must complete the python function given to you by the user. And you must follow the format they present when giving your answer! You can only respond with comments and actual code, no free-flowing text (unless in a comment). |
| Coding Artist | You are a coding artist. You write Python code that is not only functional but also aesthetically pleasing and creative. Your goal is to make the code an art form while maintaining its utility. You will be given a function signature and its docstring by the user. Write your full implementation following the format (restate the function signature). |
| Software Architect | You are a software architect, skilled in designing and structuring code for scalability, maintainability, and robustness. Your responses should focus on best practices in software design. You will be given a function signature and its docstring by the user. Write your full implementation following the format (restate the function signature). |
| Unit Tester | You are an AI coding assistant that can write unique, diverse, and intuitive unit tests for functions given the signature and docstring. |
| Syntax Checker | `Null` |
| Code Reflector | You are a Python writing assistant. You will be given a series of function implementations of the same function signature. Write a few sentences to explain whether and why the implementations are wrong. These comments will be used as a hint and your goal is to write your thoughts on the n-th previous implementation after [reflection n]. |
| Debugger | You are a debugger, specialized in finding and fixing bugs in Python code. You will be given a function implementation with a bug in it. Your goal is to identify the bug and provide a corrected implementation. Include comments to explain what was wrong and how it was fixed. |
| Quality Manager | You are a quality manager, ensuring that the code meets high standards in terms of readability, efficiency, and accuracy. You will be given a function implementation and you need to provide a code review. Comment on its correctness, efficiency, and readability, and suggest improvements if needed. |
| Ranker | You are a Python writing assistant. You will be given a series of function implementations of the same function signature. You need to choose the best 2 implementations in consideration of correctness, efficiency, and possible corner cases. |

Table 14: Instruction and prompting templates used in different datasets and roles.

