# OpenReview forum: "Dynamic LLM-Agent Network: An LLM-agent Collaboration Framework with Agent Team Optimization"
_ICLR.cc/2024/Conference — Submitted to ICLR 2024_

### Official Review · Reviewer_yyiU · 2023-10-25

**Soundness:** 3 good
**Presentation:** 2 fair
**Contribution:** 2 fair
**Rating:** 5
**Confidence:** 3

**Summary:**

The paper proposed a feedforward dynamic LLM agent network which can collaborate to improve the performance on reasoning and coding tasks. Previous work often include a static set of agents, which does not generalise to various tasks and requires strong human priors. In contrast, DyLAN (Dynamic LLM-Agent Network) has the following advantages with their design: 1. agents with different roles can exchange messages via the feedforward network 2. early-stopping when the agents reach consensus 3. automatic team optimisation by propagation (rating their predecessors in the network), aggregation (aggregate ratings from successors to quantify an agent's contribution) and selection (selecting top performing agents according to their scores derived from the propagation and aggregation steps). Finally, the model is evaluated on reasoning and coding tasks and demonstrates improved performance with reasonable computation cost.

**Strengths:**

- The paper is written clearly and is easy to follow. The comparison with baseline methods are clearly illustrated in table 1.
- The design of the feedforward communication and dynamic optimisation structure seems straightforward to implement and easily generalisable to various different types of tasks which require multi-agent collaboration and does not require strong human prior.
- The case study (Figure 5) is helpful to understanding practical use cases of the model, and draw a connection of the proposed multi-agent feedforward interaction network to real-world software development scenarios where human developers are assigned different roles to collaborate in improving code quality.

**Weaknesses:**

One limitation, which seems to be also shared with the compared baselines, is the performance gain compared with computation cost increase, especially when compared with the single execution. For example, in table 2-4, the Overall performance improved 4% but required 7 API calls (MATH dataset), 4 API calls (MMLU dataset) and 15 API calls (HumanEval dataset) compared to 1 API call with the single execution baseline.

**Questions:**

For the coding case study, I think it is clear to me why such a multi-agent collaboration is helpful in improving the code performance. However, in the general reasoning task provided (Figure 6), it is unclear to me why it would require the language model to act 4 different roles? I see that on some topic which is debatable, perspectives from agents with diverse roles would help, but it is unclear if that is necessary on some topics which has a single correct solution?
- If an agent is assigned the role of a doctor for solving the example mathematics problem, is the underlying language model deliberately trying to act as if it doesn't know how to solve the problem despite the fact that the same language model underlies the role of the mathematician?
- For example, if we started with a majority of non-experts and a minority of experts, could the system potentially produce a wrong solution due to the non-experts reaching a consensus on a wrong solution and being the majority of the multi-agent system?

On Page 1, in the last paragraph, what does sensitivity mean?

---

> ### Author Response · Authors · 2023-11-21
>
> We thank the reviewer for the constructive comments. Below, we provide detailed responses to each of your points.
>
> 1. **Performance Gain vs. Computation Cost:**
>
> We acknowledge your observation regarding the performance gain relative to the computation cost increase in experimental results. We want to clarify that **current multi-agent methods inherently involve a trade-off between performance gains and computational costs** and were discussed limitedly in previous works. Instead, we take it as an important factor.
>
> It is important to note that our methods are orthogonal to different prompting methods, as shown in Table 2. **While prompt engineering methods can improve performance without increasing #API calls, our framework demonstrates improvement across different prompting methods by leveraging the collaborative strengths of multiple agents.**
>
> Compared with single execution, the seemingly marginal improvement from **all current methods** actually results from the limited collaborative capacity of the backbone model. We still view the LLM-agent collaboration as a promising direction for better utilization of LLMs. And we thought of two possible approaches to address it in the future work: (1) build LLMs that inherently incorporate collaboration mechanisms; (2) investigate specific scenarios that single LLMs struggle to complete, e.g., agents have exclusive resources or are distributedly deployed.
>
> 2. **Necessity of Multi-Agent Collaboration:**
>
> Please refer to General Response 2 for overall responses to your concern about the collaborations of agents with different roles. Here're some detailed explanations:
>
> Your question about the role of a doctor in solving a mathematics problem is insightful. There are chances that agents with certain roles refuse to answer specific queries. Generically, agents' behaviors are unpredictable without their actual responses, indicating that ideal agent team optimization could not be achieved based on human priors (also see General Response 5). **Since different roles provide diversity as well as potential poor performance, we propose a posterior agent team optimization method in DyLAN to mitigate this gap.**
>
> Specific to your concern, as we described in General Response 2, Doctor, Programmer, Economist, and Mathematician agents may exclusively have correct answers to different mathematical queries, and it's difficult to distinguish whose answer is the correct one. Moreover, they may all have false answers but come up with a correct one after a discussion on the reasoning process / factual knowledge / probable hallucination. In conclusion, given that we couldn't predict whether an agent will respond faithfully according to its backbone model, give better solutions due to the stimuli of tools and role prompts, or even refuse to answer, **collaboration is necessary and is usually (if not always) better for accuracy on some topics even if they have a single correct solution**.
>
> We will clarify this point by stating the discussion explicitly and demonstrating clearer cases in the case study.
>
> 3. **Risk of Non-Expert Consensus Leading to Incorrect Solutions:**
>
> We want to clarify that in most subjects of MMLU, there are only 1-2 expert agents among seven candidates in our experiments, and no expert agents for a few subjects, aka. **we have already conducted experiments with a majority of non-expert agents**. Please refer to General Response 3 for further discussion about the imbalance of experts and non-experts.
>
> Regarding the potential risk of non-experts reaching a consensus on an incorrect solution, **our Agent Team Optimization algorithm plays a key role in mitigating this**, as described in General Response 3. In extreme cases, where all agents are designed to perform poorly in specific tasks, possible techniques could be designed in addition to agent team optimization, such as automatically creating agents before optimization with validation, which is not our primary focus.
>
> 4. **Clarification of "Sensitivity":**
>
> In the last paragraph of Page 1, the term "sensitivity" refers to the system's low adaptability to the domain and complexity of different tasks. In a static setup, the lack of flexibility can lead to suboptimal performance when faced with tasks that deviate from the expected domain or complexity. Our dynamic multi-round approach in DyLAN and Agent Team Optimization method is designed to adapt more effectively to a broader range of tasks. We will clarify the term in the updated version.

---

> > ### Comment · Reviewer_yyiU · 2023-11-22
> > **thank the authors for their rebuttal**
> >
> > thank the authors for their rebuttal and I have no further questions.

---

### Official Review · Reviewer_9MJA · 2023-10-31

**Soundness:** 3 good
**Presentation:** 4 excellent
**Contribution:** 3 good
**Rating:** 6
**Confidence:** 3

**Summary:**

This paper introduces the Dynamic LLM-Agent Network (DyLAN), a framework for improving LLM agents collaboration. DyLAN structures the interaction between agents in a feed-forward manner, with an early-exit mechanism and inference time agent selection. It also provides a 3-step agent optimization algorithm (with a metric called Agent Important Score). The framework is evaluated on a number of tasks and has shown performance improvement over a single agent.

**Strengths:**

- As far as I know, formulating the LLM agent interaction as a feed-forward network seems novel.
- The proposed method agent selection method and early exit method seem to improve performance and reduce the number of communication rounds.

**Weaknesses:**

- It would be beneficial if some of the details of the method were discussed more clearly. For example, how are the agents different from each other? Did you manually create a list of agents before running the experiments?
- It seems the agent team optimization part is an additional step to the inference, how many rounds of communication and "training" are required to get a specific task ready for inference with high accuracy? Do you need like 10% or more of the dataset for this optimization before you can use it for inference?

**Questions:**

- When top k agents are selected for the inference agent selection, how is the k selected?
- Are the prompts the same between the agents when queried? Some of the existing methods assign fixed roles, which will guide the LLM to specialize in certain tasks/subtasks, how is the proposed method able to do that?
- In the evaluation section, for Table 4, it seems the gain from your method to the baseline is smaller when using GPT-4 (than GPT3.5). Will this method still be effective when the single LLM model performance improves?

---

> ### Author Response · Authors · 2023-11-21
>
> We thank the reviewer for the constructive feedback and comments. Below, we address each of your points in detail.
>
> 1. **Clarification on Agent Differentiation and Construction:**
>
> Please refer to General Response 4 for detailed clarification.
>
> In DyLAN, the prompts are not the same between prompt-based agents according to Section 3.1 & 4.1 and Appendix E. This approach allows the agents to specialize in specific tasks or subtasks, contributing their unique strengths to the collaborative effort and utilizing different tools in various tasks. While diverse prompts may not significantly affect performance in some tasks like MATH, this specialization is crucial for achieving high accuracy in other scenarios like general reasoning and code generation.
>
> Specific to your concern, agents with specific expertise could be automatically settled in each time step by the Agent Team Optimization process. Orthogonally, agents can be manually set in different time steps; e.g., we designate four code writers and four judges in different time steps for code generation tasks, as exhibited in Figure 5. Thus, agents could collaborate at both task and subtask levels.
>
> Our revised paper will provide a more precise explanation of the process and construction of agent teams.
>
> 2. **Agent Team Optimization Process:**
>
> The agent team optimization in DyLAN is indeed an additional step. Please note that the process is unsupervised, meaning it does not rely on labeled data. As described in Appendix C.1, the additional consumption of API calls is 8.30 on average for general reasoning tasks and 23.04 on average for code generation. We did not add these costs to the collaboration process of optimized teams of agents because the unsupervised optimization could be performed offline, and the scores are reusable. Moreover, according to the first paragraph of Section 4.3, only a small amount of data is enough for optimization towards higher accuracy.
>
> Also, the unsupervised optimization process required no extra data as an additional step to inference. Please note that it is not an overfitting since no label is used during the process. We will emphasize these details in our revised paper to provide a comprehensive understanding of the agent team optimization process.
>
> 3. **Selection of Top k Agents and Efficiency:**
>
> k is a super-parameter that is tuned to balance efficiency and performance. We assign k=2 because it is the minimal number for agents’ interactions, and empirically found it brings a great trade-off between effectiveness and efficiency. We will include the information on how k is selected in our revised paper.
>
> 4. **Performance Improvement Relative to Backbone Models:**
>
> Regarding the observation in Table 5 about the smaller gains over the baseline when using GPT-4 compared to GPT-3.5, we believe this is due to the simplicity of HumanEval benchmark instead of the ineffectiveness of the method, since the score is 90+ and the improvement seems to be marginal. It is easy to understand that very weak LMs cannot collaborate and solve tasks, but DyLAN with GPT-3.5 actually gains excellent improvement. Thus, **the improvement of backbone models actually makes collaboration and agent team optimization more effective**. That is because capacities of reasoning, instruction following, etc., determine the effectiveness of LLM-agent collaborations.
>
> As **for empirical results**, due to budget limits, we have only tested DyLAN on the humanities category of MMLU with GPT-4-0314, and we found an improvement from 83.9 (single execution) to 87.9 (+4.0), greater than +2.3 (from 59.8 to 62.1) with GPT-3.5 in Table 3. Nonetheless, we acknowledge that, ideally, if single agents are capable of solving tasks in whatever complexity at 100% accuracy, DyLAN will be meaningless. Nevertheless, in fact, even GPT-4 can only reach <50% accuracy or even lower in some challenging tasks, including MATH (we have tested), WebArena [1], SWE-bench [2], etc., denoting strong potential for LLM-agent collaboration and agent team optimization methods in DyLAN.
>
> We are also experimenting with how our framework can be adapted to leverage the collaborative potential of LLMs in a more challenging task - WebShop. If time permits, we will also update the results.
>
> [1] WebArena: A Realistic Web Environment for Building Autonomous Agents. https://arxiv.org/abs/2307.13854
> [2] SWE-bench: Can Language Models Resolve Real-World GitHub Issues? https://arxiv.org/abs/2310.06770

---

### Official Review · Reviewer_qY2A · 2023-10-31

**Soundness:** 3 good
**Presentation:** 3 good
**Contribution:** 2 fair
**Rating:** 6
**Confidence:** 3

**Summary:**

This paper presents Dynamic LLM-Agent Network (DyLAN) for LLM-agent collaboration on complicated tasks like reasoning and code generation. It improves the efficiency and the performance of LLM-agent collaboration via inference-time agent selection at a middle-time step and byzantine consensus to terminate inference at a proper layer. They further design an automatic agent team optimization algorithm to optimize the composition of agents for DyLAN based on an unsupervised metric Agent Importance Score,

**Strengths:**

1. DyLAN properly combines multiple standard techniques to improve the performance and efficiency of LLM-agent collaboration on complicated tasks.
2. In the ablation studies section, the paper empirically shows how different strategies can individually and together contribute to improvement.

**Weaknesses:**

1. Novelty is somewhat limited since the framework is primarily a combination of standard technologies.
2. The baseline (i.e., random selection) for agent team optimization is very weak.

**Questions:**

In addition to empirical results, do you have any insights to explain why Shapley Value is not a good metric?

---

> ### Author Response · Authors · 2023-11-21
>
> We thank the reviewer for the constructive comments. Below, we address each of your concerns in detail.
>
> 1. **Novelty of the Framework:**
>
> Please refer to General Response 1 for the overall explanation of our contributions. We understand your perspective on the perceived limited novelty of our framework, given its use of standard technologies. However, we argue that the novelty of our work lies in **the unique concept of agent team optimization, and the innovative and effective method based on the unsupervised metric Agent Importance Score**. We will emphasize and clarify the point in the updated version of our paper.
>
> 2. **Baseline for Agent Team Optimization:**
>
> Please refer to General Response 5 for the additional baseline (Human Prior Selection) for Agent Team Optimization. We demonstrated that the current agent team optimization method is posterior and based on actual interactions and peer ratings between agents. **It exhibits significant improvement compared to prior selections.** As a detailed explanation, prior selections seem to overweigh the statement from role prompts and fail to predict each agent's actual behaviors and the functions of tools from short descriptions. We will add the results and the discussion in the updated paper.
>
> 3. **Shapley Value as a Metric for Individual Contributions:**
>
> Shapley Value is **well-used for measuring marginal (individual) contributions in multi-agent settings**. We chose not to use Shapley Value as a metric for agent team optimization because it is **inherently a supervised metric**, requiring labeled data to accurately assess the contribution of each agent, let alone its **high computational costs**. It's impractical in real scenarios and limited in generalizability. In contrast, our Agent Importance Score is an unsupervised metric designed to evaluate agent contributions without needing labeled data. This approach is more aligned with the practical constraints and diverse applications of LLM-agent collaborations, where labeled data may not always be available or feasible to obtain. Appendix D.5 elaborates on **Shapley Value as the self-evident metric for validating Agent Importance Score**. We will emphasize this point in the updated version of the paper.

---

### Official Review · Reviewer_KzWB · 2023-10-31

**Soundness:** 3 good
**Presentation:** 2 fair
**Contribution:** 3 good
**Rating:** 8
**Confidence:** 2

**Summary:**

This paper proposed an approach called DyLAN to optimize the performance of an ensemble of LLMs when answering a query of interest. DyLAN enables better collective performance by having a collaboration framework where ensemble members interact for multiple iterations and rate each others' responses to the query. DyLAN then proposed (i) an **agent selection mechanism** that filters out a few ensemble members with the worst responses and (ii) an **early stopping mechanism** that stops interaction between ensemble members once 2/3 of the ensemble members agree on a common response.

The authors demonstrate that DyLAN achieves better accuracy than baselines in three tasks requiring the ensemble to reason or generate code. DyLAN was compared against representative works that also attempt to improve the performance of a collection of interacting LLMs. Ablation studies were also conducted to demonstrate the importance of various factors related to DyLAN's training process and its own agent selection and early-stopping mechanism.

**Strengths:**

**Minor Strength - Originality - Method Novelty**

To the best of my limited knowledge about training an ensemble of LLMs, DyLAN seems to be a novel approach. At least compared to the methods provided in the related section of the work, DyLAN appears to have significant differences (i.e. with respect to the early stopping and agent selection mechanisms) with previous work, which seem plausible in further improving the performance of an ensemble of LLMs. Nonetheless, the topic of training a collection of interacting LLMs itself seems to have been explored by previous works already.

**Minor Strength - Quality - Method Soundness**

Except for some minor weaknesses (written in the section below), DyLAN's early stopping and agent selection mechanism seems generally sound from a multiagent systems perspective. Given the existence of a sufficient number of LLMs that can expertly handle an input query, I find it plausible that DyLAN should be able to identify a smaller number of LLMs whose response we can refer to once they achieve consensus.

**Major Strength - Quality - Experiment**

While I cannot comment much on the baseline selection following my limited knowledge of the topic, I find that the designed experiments were well designed to demonstrate DyLAN's advantages in terms of its overall accuracy and incurred cost during an interaction. At the same time, I highly appreciate the ablation study conducted by the authors to investigate the importance of the early-stopping and agent selection mechanisms. I especially like how it demonstrates the importance of the agent selection mechanism.

**Major Strength - Clarity - Experiment Analysis**

As a reader with less expertise in this topic, I also highly appreciate how the authors carefully outlined the different insights gained from each experiment. This helps in pinpointing the importance of the DyLAN's various components and in gaining an understanding of the capabilities achieved by DyLAN. Similarly, insights gained from comparison against different baselines were properly written down, also making it easier for a reader with less expertise in this topic.

**Weaknesses:**

**Minor weakness - Clarity - Lack of Problem Formulation**

It was slightly tricky to grasp the type of queries being solved by the ensemble of LLMs and how they interact with each other. While it is unfortunate that it was relegated to the appendix, Figure 5 is an excellent figure that could have helped readers understand the problem being solved if it had been presented earlier. In place of Figure 5 which seems to take a lot of space, perhaps the authors could consider describing a formal model of the interaction between agents and decision-making problems just to give more context.

**Minor Weakness - Soundness - Expert LLMs being outnumbered by non-expert members of the ensemble**

I suppose one of the weaknesses of DyLAN is when the number of expert LLMs for dealing with a particular query is significantly lower than the number of non-experts. If somehow the number of non-expert LLMs selected during agent selection is still larger than the experts after agent selection, DyLAN can still yield poor accuracies. Perhaps it could be useful to have multiple rounds of agent selection until some average score/metric (ideally reflecting their capacity in solving the task) of the remaining LLMs is above a certain threshold.

**Minor Weakness - Soundness - Absence of reliable rankers**

Another possible pitfall occurs when DyLAN does not have a reliable ranker for agent selection, which results in the selection of possibly highly suboptimal members of the ensemble for the final decision.

**Questions:**

1. Is DyLAN equipped with a mechanism to deal with a severe imbalance between expert and non-expert agents?
2. Can you explain the rankers considered for DyLAN and why a specific ranker is chosen?

**Details Of Ethics Concerns:**

I have ethical concerns of the method.

---

> ### Author Response · Authors · 2023-11-21
>
> We thank the reviewer for the constructive feedback and comments. Below, we respond to each of your concerns in detail.
>
> 1. **Clarity and Problem Formulation:**
>
> We appreciate your suggestion regarding our formulations. We agree that Figure 5 is a crucial element in conveying the concept and operations of DyLAN. Currently, we demonstrate the formal model for agent interactions as a feed-forward network $S=(A, E)$, in Section 3.1, and we described how task query $q$ is solved by the system's output $o$ in the last paragraph. Based on your suggestion, we will consider including a formal model, including the decision-making process earlier in the text, to provide readers with a clearer and more immediate understanding of the problem being solved.
>
> 2. **Handling Imbalance Between Expert and Non-Expert Agents:**
>
> Your concern about the potential imbalance between expert and non-expert LLMs in DyLAN is insightful. Please refer to General Response 3 for detailed explanations. Based on empirical results, we explained that **Agent Team Optimization could deal with the imbalance between expert and non-expert agents**. Implementing multiple rounds of agent selection is likely to be effective in addressing extreme cases, as you suggested. We believe this approach will enhance the robustness of DyLAN against the risk of poor accuracies due to an imbalance in expertise.
>
> **To step further, we want to share some thoughts on building metrics that reflect an agent's capacity for certain tasks**. A new work demonstrates several aspects for evaluating agents under multi-agent settings [1]. We think it is possible to extend its metric to certain timesteps inside DyLAN for a clear understanding of agents' capacities. We will add the discussion to our updated paper.
>
> 3. **Selection and Effectiveness of Rankers:**
>
> Regarding the rankers used in DyLAN, we conducted an ablation study to evaluate the performance of various ranking methods in Appendix D.3. The choice of a specific ranker was based on its ability to improve the overall performance with reasonable computational costs. As described in the second paragraph in Section 2, there are mainly two types of rankers - listwise and pairwise. The former ranks candidates in one pass, and the latter ranks candidates by scoring them based on the comparison result of each pair of candidates. We tested three kinds of pairwise candidates, including LLM-Blender (using GPT-3.5 to judge each pair), Elo Score (using GPT-3.5 as a judge within theoretic probability distribution models), and Sliding Window (using GPT-3.5 with only $nk$ times comparison for top k), along with listwise ranker (based on GPT-3.5).
>
> We eventually found **their performance roughly the same** for DyLAN, and the listwise method saved computational costs. It might be because of the strong discrimination ability of GPT-3.5. We use the listwise ranker for all other experiments, and it indeed demonstrates effectiveness and efficiency on downstream tasks. Also, we acknowledge that multiple rankers could be potentially beneficial for DyLAN but may introduce extra costs. We will state these discussions explicitly in the updated paper.
>
> In conclusion, we believe that our paper's additional explanations, experiments, and reorganization, as suggested in response to your feedback, will significantly strengthen our work. We are committed to advancing the field of LLM-agent collaborations and appreciate your valuable input in helping us refine our approach.
>
> [1] MAgIC: Investigation of Large Language Model Powered Multi-Agent in Cognition, Adaptability, Rationality and Collaboration. https://arxiv.org/abs/2311.08562

---

> > ### Comment · Reviewer_KzWB · 2023-11-22
> > **Response to Author Comments**
> >
> > Thank you for your response. It does help address some of my concerns. As such, I will keep my original scores.

---

### Official Review · Reviewer_mH5o · 2023-11-01

**Soundness:** 2 fair
**Presentation:** 2 fair
**Contribution:** 2 fair
**Rating:** 3
**Confidence:** 3

**Summary:**

This paper suggests the creation of a tactical group of agents that communicate within a flexible interaction structure tailored to the specific task at hand. The approach involves developing a system called the Dynamic LLM-Agent Network (DyLAN), which facilitates collaboration among LLM agents on complex activities, including reasoning and generating code. DyLAN allows for multi-round interactions among agents within an adaptable framework, incorporating on-the-fly agent selection and a premature termination feature to enhance both performance and efficiency. Additionally, this paper introduces a method for the automatic optimization of the agent team, utilizing an unsupervised metric named the Agent Importance Score. This score helps in determining the most effective agents by evaluating the individual contributions of each agent.

**Strengths:**

Pros:
1. This paper proposes a framework DyLAN, which presents a novel structure for LLM-agent cooperation, assembling agents in a tiered, feed-forward network that features adaptive architecture, incorporating mechanisms for selecting agents during inference and halting the process prematurely when necessary.
2. To optimize agent collaboration within DyLAN, this paper crafted a self-governing optimization algorithm that leverages an Agent Importance Score determined without supervision, aiming for a balance of performance and efficiency.
3. This paper claimed that DyLAN has been shown to deliver strong results in accuracy, efficiency, and consistency across a spectrum of tasks, including general reasoning, numerical problem-solving, and the generation of code.

**Weaknesses:**

Cons:
1. This paper cannot demonstrate that agent collaboration must perform better than single agent. For example, in Figure 6, the agent collaboration is totally unnecessary. The roles such as Programmer, Economist and Doctor are useless for the math reasoning problem.
2. The results are not convincing. For example, does agent collaboration really perform better than single agent on math reasoning? The example in Figure 6 does not show the benefit of agent collaboration on math reasoning.
3. The collaboration process is unclear. For example, in Figure 5, how does the algorithm developer and programmer collaborate together? It seems programmer can already finish the task very well.
4. It is suggested that the authors explain more clearly the function of Agent Importance Scores using examples. Otherwise, it is unclear why it is necessary. This paper says that “We ignore the peer rating scores in responses of agents for computing Agent Importance Scores.” In Figure 5. The author may not ignore the peer rating process.
5. The contribution is limited. The agent collaboration framework only use “role-playing” for different agents, which is proposed in the previous paper [1], and also discussed in many previous papers such as [2]. It is suggested that the authors could consider adding tool use. For example, it does not make sense to use LLMs as unit tester (in Figure 5).
6. Some important references are missing. It is worth noting that this paper does not cite the paper [1] which proposed “role-playing”.

[1] CAMEL: Communicative Agents for "Mind" Exploration of Large Language Model Society NeurIPS 2023 https://arxiv.org/abs/2303.17760

[2] Unleashing cognitive synergy in large language models: A task-solving agent through multi-persona self-collaboration https://arxiv.org/abs/2307.05300

**Questions:**

See weaknesses.

---

> ### Author Response · Authors · 2023-11-21
>
> We thank the reviewer for the constructive feedback and comments. Below, we address each of your concerns in detail.
>
> 1. **Clarification on the contributions and motivation**
>
> Please refer to General Response 1 for the summary of the contributions. We want to clarify that "role-playing" agents are not our focus and are **only used partially in both the formulation and experiments**.
>
> 2. **Addressing Factual Errors:**
>
> We acknowledge the confusion caused by certain aspects of our presentation. Here are clarifications:
>
> 2.1. Empirically, we have compared the performance of **single and multiple agents** (in Table 2, 3, and 5), and our findings align with previous research underscoring the benefits of multi-agent collaboration.
>
> 2.2. In response to your suggestion regarding **tool usage**, we refer to our General Response 4, where we elaborate on tool usage in DyLAN, as described in Section 3 & 4.1.
>
> 2.3. In Figure 5, **Unit Tester** is indeed an LLM agent equipped with Code Interpreter as a tool, according to the last sentence of Section 4.1 and the last paragraph of Appendix C.1. We didn't deploy extra tools except code interpreters because of the alignment towards baseline methods.
>
> 2.4. We ignore the scoring process in Figure 5 mainly due to simplicity since there are 8 agents in Figure 5. We chose to demonstrate **the scoring process in Figure 6** with only 4 agents.
>
> We will clarify these points and rectify any wording issues in our revised paper.
>
> 3. **Necessity and Performance of Agent Collaboration vs. Single Agent:**
>
> Please refer to General Response 2 for the discussion of the effectiveness of LLM-agent collaborations. Detailed explanations about comparing the collaboration with single LLM agents are as follows.
>
> Our work is grounded in the philosophy that while a single, perfect agent capable of efficiently handling all tasks is an ideal goal, it is not yet achievable. In this context, our Dynamic LLM-Agent Network (DyLAN) framework aims to leverage the strengths of multiple agents in a collaborative setting to achieve better results and cost-efficiency, as well as better generalizability compared to a single agent.
>
> From our observations in current experiment settings: (1) Expert is hard to manually select when there are many candidates; (2) The selection from human priors (by LLMs) mismatches the optimal team of agents and cause severe performance drop (refer to General Response 5).
>
> In response to your comments, **Mathematician and Programmer agents alone are not enough for all mathematical and programming problems, respectively**. For instance, Programmer and Economist agent can outperform Mathematician agent in elementary mathematics, and a single programmer in Figure 5 (based on GPT-35-turbo-0301) could only achieve 73.2 on HumanEval compared to 82.9 from DyLAN. Collaboration. We'll add a different case to demonstrate the effectiveness of collaborations, e.g., DyLAN solves a query where all single agents fail.
>
> 4. **Clarification of the Collaboration Process:**
>
> To be specific, in Figure 5, Programmer and Algorithm Developer agents write incorrect programs at t=1, and at t=3, they receive the judgment from judges at t=2, thereby perceiving other cases along with according judgments from Unit Tester, Syntax Checker, and other judges. Programmer at t=3 refined its solution, which turned out to be correct, while Algorithm Developer still failed. The process demonstrates that code writers refine their solutions based on the feedback from judges and ensemble cases from other code writers. After Inference-Time Agent Selection, the other two agents reach a consensus and vote for the final, correct answer.
>
> In short, **the interplay of agents' distinct expertise leads to a more robust and accurate solution than what a single agent could achieve**. We will add these descriptions to the caption of Figure 5. Additionally, to demonstrate the necessity and effectiveness of Agent Importance Score in Figure 5, we will include more detailed examples in the revised version.
>
> 5. **Missing References:**
>
> We apologize for the oversight in not citing the relevant paper. Though "role-playing" is not our core focus, we acknowledge its importance in the field of LLM agents. The reference will be duly added in the next version of our paper.

---

### Official Review · Reviewer_CxXZ · 2023-11-07

**Soundness:** 3 good
**Presentation:** 3 good
**Contribution:** 2 fair
**Rating:** 5
**Confidence:** 3

**Summary:**

The main contributions of this paper are:

- Proposing a new framework called Dynamic LLM-Agent Network (DyLAN) for organizing collaboration between multiple LLM agents. DyLAN allows agents to interact over multiple rounds in a dynamic architecture.
- Introducing two key components in DyLAN - inference-time agent selection to filter out low-performing agents, and an early stopping mechanism based on Byzantine consensus theory to improve efficiency.
- Developing an unsupervised metric called Agent Importance Score to quantify each agent's contribution, which can then be used to automatically optimize the composition of the agent team for a given task.
- Demonstrating DyLAN's effectiveness on multiple tasks including reasoning, arithmetic and code generation, showing accuracy improvements and reasonable computational cost compared to baselines. Agent team optimization is shown to further boost performance on specific tasks/domains.

**Strengths:**

- The technical claims around the DyLAN framework and agent team optimization seem reasonably sound, though not groundbreaking. The core ideas build incrementally on related work.
- The experimental methodology is generally solid, with evaluations on multiple representative tasks using reasonable baselines. However, comparisons to some very latest multi-agent LLM methods are lacking.
- The central claims around DyLAN's performance are supported reasonably well by the experiments. The gains over baselines help validate the techniques.
- Analysis and insight could be deeper around why agent team optimization works and whether the proposed scoring method effectively captures agent contributions.
Overall the paper quality seems good. The techniques seem to work fairly well in practice. But the novelty and advancement over recent related work appears modest.

**Weaknesses:**

The core ideas around dynamically organizing LLM agent collaborations and automatically optimizing the agent team seem novel and could be of interest to the ICLR community. However, the paper is generally incremental work building on a lot of recent research around multi-agent LLMs. The techniques used, like inference-time pruning and unsupervised contribution scoring, are not completely new concepts either.
In more details:
- The experimental validation is reasonable but mainly incremental - evaluating on established datasets with existing baseline methods. More complex, real-world tasks could be illustrative.
- The paper claims efficient collaboration but the overhead of techniques like peer rating and consensus checks could be significant. More analysis on computational costs is needed.
- There is limited ablation study or analysis into the agent scoring method. It is unclear if it is actually capturing meaningful contribution effectively.
- Why was CodeT chosen as the baseline? This does not appear to be a reasonable baseline. CodeT is not a multi-agent framework for code generation. It works on the principle of generating more test cases in order to improve the ranking of the solutions. Stronger baselines should be chosen for a fairer comparison.
- The presentation seems repetitive in parts with previous sections being paraphrased. More clarity and conciseness in writing would strengthen the paper.

**Questions:**

- How does DyLAN's performance compare to other very recent works on multi-agent LLMs? The baselines used seem a bit outdated.
- Can you provide more analysis/insight into why agent team optimization works well? Is the Agent Importance Score capturing something meaningful?
- Have you experimented with DyLAN on more complex, open-ended tasks beyond the datasets used? How does it perform?
- Could DyLAN be extended to do online agent team optimization during inference as well?
- For early stopping via Byzantine consensus, how did you determine the optimal threshold for answer similarity? Was this tuned per task?

---

> ### Author Response · Authors · 2023-11-21
> **Individual Response 1**
>
> We thank the reviewer for the constructive feedback and comments. Below, we respond to each point in detail.
>
> 1. **Summary of Key Contributions and Novelty:**
>
> Please refer to General Response 1 for the summary of contributions. Our core contribution of improving LLM-agent collaborations by optimizing agent teams is **distinct from previous methods or concepts**, especially as we proposed an unsupervised metric Agent Importance Score to implement the posterior optimization.
>
> 2. **Experimental Validation and Real-World Applicability:**
>
> Our experiments were designed to provide a clear and objective evaluation of DyLAN's performance. We chose established general datasets and **the strongest baselines (before the submission time)** for their recognized validity in the research community. All baselines are chosen for fair reasons. Specifically, CodeT is the strongest method based on GPT-35-turbo before submission. Even though Reflexion managed to reach SOTA using GPT-4, it still performs worse than CodeT on GPT-35-turbo, as shown in Table 5.
>
> We acknowledge that CodeT is not a multi-agent method, but **it serves as a strong baseline even with multi-agent methods coming out near the submission date**. Because we were not able to re-implement all of them, we post a few reported results from recent (concurrent) multi-agent works on HumanEval with different backbone models:
>
> |Method|GPT-35-turbo|GPT-4|
> |:----|:---:|:---:|
> |AgentVerse [1]|75.6|89.0|
> |CAMEL [2]|69.4| - |
> |MetaGPT [3]| - |85.9|
> |The best baselines in our paper|74.8 (CodeT)|91.4 (Reflexion)|
> |**DyLAN** (*Ours*)|**82.9**|**92.1**|
>
> Regarding MMLU and MATH datasets, most multi-agent works [4] have not reported their results on the datasets or used the same backbone model, and it’s out of budget for us to reimplement those methods on GPT-35-turbo. Nonetheless, we demonstrate our fair reasons for baseline selections and the effectiveness of DyLAN.
>
> We acknowledge your point about the potential benefits of evaluating DyLAN on more complex, real-world tasks. As part of the rebuttal response, we are extending our evaluation to more diverse and challenging scenarios, such as WebShop [5], to demonstrate the applicability of our framework further. Due to the time limit, we plan to update the results later.
>
> 3. **Clarification on Computational Overhead and Efficiency:**
>
> Regarding the computational overhead of techniques like peer rating and consensus checks, we have **already included these factors** in the number of API calls reported in our experiments:
>
> - In Appendix C.2, we mentioned that the rating is implemented by the suffix of prompts. Thus, the rating process does not affect #API calls. Indeed, this kind of prompting would increase token consumption. However, we find it insignificant since CoT prompts, reasoning processes, mathematical formulas, and codes take up most of the token consumption.
>
> - In the last two paragraphs of Appendix C.1, we mentioned that consensus checks are processed by template matching with pure scripts. Thus, the overhead is small and does not change #API calls as well.
>
> We appreciate your concern about our potential mistakes. We will explicitly describe the overhead above in the next version of the paper.
>
> 4. **Clarity and Conciseness of Presentation:**
>
> We appreciate your feedback on our paper's presentation. We will undertake a thorough review to eliminate repetitive content and improve clarity and conciseness.
>
> [1] AgentVerse: Facilitating Multi-Agent Collaboration and Exploring Emergent Behaviors. https://arxiv.org/abs/2308.10848
> [2] CAMEL: Communicative Agents for "Mind" Exploration of Large Language Model Society. In proceedings of the 37th Conference on Neural Information Processing Systems. https://arxiv.org/abs/2303.17760
> [3] MetaGPT: Meta Programming for A Multi-Agent Collaborative Framework. https://arxiv.org/abs/2308.00352v5
> [4] AutoGen: Enabling Next-Gen LLM Applications via Multi-Agent Conversation. https://arxiv.org/abs/2308.08155
> [5] WebShop: Towards Scalable Real-World Web Interaction with Grounded Language Agents. https://arxiv.org/abs/2207.01206

---

> ### Author Response · Authors · 2023-11-21
> **Individual Response 2**
>
> 5. **Analysis and Insights on Agent Team Optimization:**
>
> The effectiveness of our agent team optimization approach is a key contribution of our work. The Agent Importance Score is designed to capture the contributions of individual agents within a collaboration based on actual responses.
>
> First, we want to clarify the **motivation** of the scoring method. The scoring method is inspired by neural Importance Score in traditional machine learning studies, as explained in Section 2. While the algorithm couldn’t be directly transferred, we propose a three-step algorithm (as described in Section 3.3) by leveraging peer ratings to achieve effectiveness and efficiency. There are also several human-team optimization findings supporting our methodology, as shown in Section 2.
>
> We also **provided empirical results to demonstrate Agent Importance Score as the indicator of individual contributions**. (1) The direct proof is the performance improvement after optimization in downstream tasks. In the last paragraph in Section 4.2 and the second paragraph in Section 4.3, we clearly describe the benefits of Agent Team Optimization and the impact of team sizes. (2) We also compared the scoring distribution with Shapley Value (Appendix D.5). We found that Agent Importance Score has a closer distribution to Shapley Value, which shows that the scoring process grasps the individual contributions of each agent to a fair extent.
>
> Some insights could be derived from the existing results. As shown in Table 6, Agent Team Optimization is capable of **differentiating contributory agents** from the others. Some contributory agents match human prior, e.g., Mathematician for college mathematics, and some seem to be unrelated or not directly related, e.g., Psychologist for public relations. It reveals that Agent Team Optimization could not only retrieve domain experts but might **also retrieve agents with other domain knowledge that might help** (refer to Appendix D.1 & D.4). Moreover, peer ratings are more reliable than self-confidence due to previous studies [6], which provides extra solid support for our method.
>
> Please refer to General Response 5 for the additional experiment of another baseline for Agent Team Optimization. We will add the discussion in the next version of our paper.
>
> 6. **Extension to Online Agent Team Optimization:**
>
> First, we want to clarify that **Agent Team Optimization does not depend on the label or the ground truth of datasets**. The optimization is based on the unsupervised metric Agent Importance Score. Thus, the optimization method **can already be performed posteriorly during inference** at this stage.
>
> Regarding online agent team optimization methods, DyLAN currently employs Inference-Time Agent Selection as a naive approach of online optimization, as described in Appendix B (Limitation). **The challenge is to capture the actual multi-round behaviors of agents in the middle of collaboration**. Inference-Time Agent Selection only considers single-round responses and does not consider the ratings from the agents themselves. There is potential for further development in this area, allowing for more concrete and adaptive online agent team optimization.
>
> 7. **Optimal Threshold of Answer Similarity for Early Stopping:**
>
> In Appendix C.1, we described using exact match for classification queries and BLEU scores for open-ended generation tasks, including code generation tasks. We set the threshold of BLEU score to 0.9 empirically for all open-ended generation tasks without specific tuning since it’s not our primary focus. In Appendix B (Limitation), we acknowledge that the threshold and choices of methods could be further optimized, and based on our observation, it does affect the performance, though marginally.
>
> [6] Can LLMs Express Their Uncertainty? An Empirical Evaluation of Confidence Elicitation in LLMs. https://arxiv.org/abs/2306.13063

---

### Author Response · Authors · 2023-11-21
**General Response (1)**

We sincerely appreciate the valuable feedback from all reviewers and will carefully incorporate it into our paper. We respond here to address shared concerns. Please refer to individual responses for detailed explanations.

1. **Summary of Key Contributions and Motivations (mainly for reviewers CxXZ, mH5o, and qY2A)**

Our primary contribution in this work is **the concept that agent teams require optimization for effective and efficient LLM-agent collaborations**. To address this, we introduce a general framework - Dynamic LLM-Agent Network (DyLAN). **The innovative team optimization algorithm and the methodology of DyLAN** collectively enhance the performance of agent collaborations across various tasks. Here are the key contributions:

1.1. **Emphasis on the Necessity of Agent Team Optimization:**

We assert that optimizing agent teams is crucial for advancing LLM-agent collaborations. Recently, LLM-generated agents have been used in collaborations [1-6]. However, these methods are restricted to producing prompt-based agents, and the generation based on short descriptions of tasks or candidate agents also embeds strong human priors. It does not consider the real behaviors of agents. Our work addresses these gaps by proposing a general framework that dynamically optimizes the agent team's composition and interaction architecture in response to specific task queries.

1.2. **Introduction of a General Framework and Team Optimization Algorithm:**

To realize optimized agent teams on various tasks, we first provide a task-agnostic formulation in Section 3.1. Based on the formulation, we implement DyLAN as a more flexible and adaptable system (Section 3.2), capable of adjusting to a wide range of tasks without the constraints of fixed structures (better E). Also, building on this formulation, we introduce an algorithm for agent team optimization (Section 3.3), enabling the selection of the most effective agent combinations (better A) based on task queries. Our approach prioritizes efficiency and optimizes teams of agents based on actual multi-round responses within the collaboration process.

1.3. **Development and Implementation of DyLAN:**

In Section 3, we detail the implementation of our solution, DyLAN. While we acknowledge that concepts like Inference-Time Agent Selection and Early-Stopping mechanisms are not entirely novel in the broader field of machine learning, their application involving high-dimensional and closed-source LMs presents challenges. Our work addresses these challenges by effectively integrating these mechanisms into the DyLAN (Section 3.2 and Appendix D.3). Furthermore, we introduce the concept of Agent Team Optimization in LLM-agent collaborations, a novel concept to the field. Prior to our work, there was no straightforward method for unsupervised measurement of language agents' contributions.

2. **Why do collaborations help? (mainly for reviewers mH5o and yyiU)**

First of all, we want to clarify that **previous works** [1-8] have demonstrated the effectiveness of agent collaborations on specific task-solving, and we summarized their opinions in Section 2. Collaborations are proven to be effective in different scenarios [1,2,5,7].

The effectiveness remains with DyLAN. It is crucial to recognize that while each agent in collaboration may have its expertise, this does not imply perfection in solving all problems within that domain. On the one hand, **LLMs with specific prompts may not behave exactly the same according to human prior** - Programmer and Economist agents can outperform Mathematician agents in elementary mathematics. It demonstrates that, without the actual response, it's difficult to determine the best agent for each query. On the other hand, collaborations outperform a single agent to a great extent due to several mechanisms, including voting, ensembling, iterative refinement, debate, and adversarial acting with feedback. **Individual agents might not achieve the correct solution independently, but it can be realized through collaborations.**

Our work in DyLAN enables agents with different specializations to contribute their unique insights and incorporate different collaboration mechanisms, thereby enhancing the system's overall problem-solving capability. For instance, Inference-Time Agent Selection serves as a new mechanism, Byzantine Consensus explicitly leverages voting, and ensembling and adversarial acting are implicitly embedded in the interaction architecture. Besides, Agent Importance Score is the key method to understand the contribution of each candidate agent based on actual responses, ensuring effective Agent Team Optimization.

Each of these mechanisms contributes to a richer, more nuanced problem-solving process, allowing DyLAN to tackle complex tasks that are beyond the scope of single agents. In the next version of our paper, we will explicitly state these insights and provide detailed case descriptions in response to reviewer yyiU's concerns.

---

### Author Response · Authors · 2023-11-21
**General Response (2)**

3. **How to deal with the imbalance of expert and non-expert agents? (mainly for reviewers KzWB and yyiU)**

Empirically, we found that **the ratio of expert and non-expert candidate agents seems not to be the key factor affecting Agent Team Optimization's performance**. For example, in code generation tasks, 12 candidates are almost related to code writing or reviewing. Thus, an improvement of +6.7 resulted from Agent Team Optimization (Section 4.2). However, most subjects in MMLU are only faced by 1-2 experts in the seven candidates, e.g., clinical knowledge, and a few of them have no related expert, e.g., management or public relations. There are still significant performance improvements in these tasks, according to Table 6, while college mathematics indeed has the highest improvement as there are more related roles.

We also found **Agent Team Optimization already has some ability to differentiate expert agents from non-expert agents under imbalanced scenarios. However, the expert agents in human prior may differ from the most contributory agents denoted by Agent Importance Score**, e.g., Lawyer for college mathematics. We also discussed the phenomenon in Appendix D.1. To provide empirical evidence, we substituted Lawyer with Historian and ran an additional experiment on college mathematics in MMLU, resulting in 55.0 (dropping 10.0 compared to 65.0 with the original composition). Thus, a Lawyer agent might be more contributory than a Historian in this task. It potentially reveals that we need to understand LLM agents from multiple perspectives under multi-agent collaborations rather than simply differentiate experts from non-experts. A new work also agrees with our findings [8].

We acknowledge that low performance might be caused if the candidates demonstrate poor diversity and the majority are of low performance. And we did not think of easy-going walkarounds. To tackle the imbalance of expert and non-expert agents, replicating some agents with high Agent Importance Score instead of including low-score agents seems to be a possible solution. Additionally, in extreme circumstances, we might need to automatically create agents with validation, in addition to agent team optimization.

We appreciate this inspiring point for analysis of current methods, and it opens up a direction for further improvement. We will add this discussion to our paper in the next version.

4. **Tools are incorporated in both formulation and experiments. (mainly for reviewers mH5o and 9MJA)**

As stated in Section 3.1, three types of agents are supported in DyLAN. Most agents are prompt-based agents, with the same task instruction and individual role prompts to stimulate their expertise (Appendix E). Besides prompts, we also equipped a code interpreter as an augmentation tool for "Unit Tester", and "Syntax Checker" is a pure tool for code generation tasks, as stated in Section 4.1 and Appendix C.1. The experimental results in Table 5 demonstrate their effectiveness in the collaboration and DyLAN's compatibility for various agents, compared to previous work [5,6].

Since we want to exhibit the effectiveness of the framework, we decide it's unfair to use many extra tools, e.g., a document retriever, a web browser, or a calculator in arithmetic reasoning tasks, since most baselines are not natively compatible. We'll make these descriptions clearer in the next version and consider collaborating agents with complex tools or retrieved documents to push the performance to limits in future demos.

[1] Improving Factuality and Reasoning in Language Models through Multiagent Debate. https://arxiv.org/abs/2305.14325
[2] Self-collaboration Code Generation via ChatGPT. https://arxiv.org/abs/2304.07590
[3] CAMEL: Communicative Agents for "Mind" Exploration of Large Language Model Society. In Proceedings of the 37th Conference on Neural Information Processing Systems. https://arxiv.org/abs/2303.17760
[4] Chameleon: Plug-and-Play Compositional Reasoning with Large Language Models. In Proceedings of the 37th Conference on Neural Information Processing Systems. https://arxiv.org/abs/2304.09842
[5] Unleashing cognitive synergy in large language models: A task-solving agent through multi-persona self-collaboration. https://arxiv.org/abs/2307.05300
[6] AgentVerse: Facilitating Multi-Agent Collaboration and Exploring Emergent Behaviors. https://arxiv.org/abs/2308.10848
[7] ProAgent: Building Proactive Cooperative AI with Large Language Models. https://arxiv.org/abs/2308.11339
[8] MAgIC: Investigation of Large Language Model Powered Multi-Agent in Cognition, Adaptability, Rationality and Collaboration. https://arxiv.org/abs/2311.08562

---

### Author Response · Authors · 2023-11-21
**General Response (3)**

5. **Additional results (mainly for reviewers CxXZ and qY2A)**

Regarding baselines for the current Agent Team Optimization method, we added Human Prior Selection, which queries LLMs to give a selection based on the descriptions of candidates and the task. Due to the time limit and the budget, we tested DyLAN with GPT-35-turbo-0301 on five subjects of MMLU and HumanEval. Here’re results:

- MMLU

| Agent Team Size |Optimization Method |College Mathematics |Management | High School Statistics| Clinical Knowledge | Public Relations |Overall |
|:---:|:---|:---:|:---:|:---:|:---:|:---:|:---:|
|7 |*(befor optimization)*|40.0|76.2|65.1|69.8|54.5|63.5 (+0.0)|
|4 | *Random Selection* |45.0|71.4|67.4|71.7|54.5|64.8 (+1.3)|
||*Human Prior Selection*|60.0|80.1|65.1|69.8|54.5|66.7 (+3.2)|
||*Agent Importance Score*|**65.0**|**90.5**|**74.4**|**75.5**|**59.1**|**73.6** (**+10.1**)|

- HumanEval

| Agent Team Size |Optimization Method | Pass@1 | #API |
|:---:|:---|:---:|:---:|
|12|*(befor optimization)*|76.2 (+0.0)|23.04|
|8 | *Random Selection* |75.6 (-0.6)|17.73|
||*Human Prior Selection*|78.0 (+1.8)|16.37|
||*Agent Importance Score*|**82.9 (+6.7)**|**16.85**|

The result shows that the current implementation with Agent Importance Score steadily outperforms Human Prior Selection. There are two major reasons: (1) **Compared to posterior optimization methods, prior selection may not grasp the actual behaviors of agents** and may not understand which agents are most contributory or helpful to others in the real collaboration process. Thus, in High School Statistics, Clinical Knowledge, and Public Relations subjects of MMLU dataset, prior selection performs even worse than random selection. (2) **Human Prior Selection may hardly select tool-augmented agents without actual analyses from peer agents.** From observation, Unit Tester and Syntax Checker were not selected for code generation tasks, which may cause lower performance.

We are also adding WebShop as an open-ended task to demonstrate the generalization of DyLAN. Due to time limits, we plan to update the result later.

6. **Practical Impact of Agent Team Optimization (for all reviewers)**

We believe the concept of Agent Team Optimization, which is central to our work, has far-reaching implications for both the technological landscape and society. In an era where a plethora of agents, including GPTs from OpenAI, are rapidly emerging, the ability to efficiently and effectively select the right agents for specific tasks becomes increasingly crucial. This optimization goes beyond mere technological advancement; it has the potential to significantly impact everyday life and various professional domains:

1. **Personalized Digital Assistants**: Imagine a scenario where digital assistants, powered by optimized agent teams, can provide more accurate, context-aware, and personalized responses to users. This could make digital assistants more like personal aides that understand and anticipate needs based on the situation, and agents exchanged between users could automatically settle into tasks where they could give high contributions.

2. **Enhanced Learning and Education**: Optimized agent teams could provide tailored tutoring in educational settings, adapting to a student's learning style and current knowledge level. This could democratize education, providing high-quality, personalized learning experiences to students regardless of geographical location or socio-economic status.
……

In conclusion, Agent Team Optimization stands at the forefront of a paradigm shift in how we interact with AI agents. Its practical applications are vast and varied, promising not only to enhance efficiency and productivity across multiple sectors but also to enrich the quality of life and foster innovation in ways we are just beginning to imagine.

---

### Public Comment · ~Edward_Y_Chang1 · 2024-05-24
**Prior work exists**

The work of SocraSynth, first published in July 2023, can be discussed and and compared in the related work section.  SocraSynth, a multi-LLM debate framework has had a 11-chapter book published in 2024.

https://www.researchgate.net/publication/374753069_Examining_GPT-4's_Capabilities_and_Enhancement_with_SocraSynth

---

### Meta-Review · Area_Chair_5f7d · 2023-12-07

**Metareview:**

In this paper, the authors introduced a new approach called DyLAN (Dynamic LLM-Agent Network), consisting of a set of LLM-based agents collaborating on complex tasks like reasoning and code generation. The approach includes multiple rounds of agent interaction, an agent selection mechanism to optimize the agent team with an unsupervised metric, and early-stopping criteria. The authors applied this approach to MATH, HumanEval, and MMLU benchmarks.

The reviewers appreciated the novelty of this approach in formulating agent selection as a feedforward communication/ interaction network and the dynamic optimization of the agent team. However, there are still some concerns in the current version of the paper, including (1) limited novelty as compared to the related multi-agent research work and the lack of experimental results to showcase the advantage of DyLAN over related agent-based SoTA baselines; (2) Limited clarity of the methods, including the description of agent definition/ differentiation, reasons why certain roles are selected, and how the agents interact with each other. I recommend the authors review all the feedback from reviewers and improve the paper accordingly. One more area the authors should investigate further is to extend their method beyond GPT-based models, such as open-source/smaller-size models, to demonstrate the generalization of their approach.

**Justification For Why Not Higher Score:**

The current paper has major limitations in terms of experimental results to compare with related multi-agent SoTA baselines. The description of the method also needs major revision to elaborate the definition of LLM agents, why certain roles are selected for the agents, and how agents can interact with each other. Many details are still unclear or only mentioned in a small part of the appendix section.

**Justification For Why Not Lower Score:**

N/A

---

### Decision · Program_Chairs · 2024-01-16

Reject